# Ten-eleven translocation 2 interacts with forkhead box O3 and regulates adult neurogenesis

Xuekun Li[1,*], Bing Yao[2,*], Li Chen[3], Yunhee Kang[2], Yujing Li[2], Ying Cheng[2], Liping Li[1], Li Lin[4], Zhiqin Wang[2], Mengli Wang[2], Feng Pan[5], Qing Dai[6], Wei Zhang[7], Hao Wu[3], Qiang Shu[1], Zhaohui Qin[3], Chuan He[6], Mingjiang Xu[5] & Peng Jin[1,2]

Emerging evidence suggests that active DNA demethylation machinery plays important epigenetic roles in mammalian adult neurogenesis; however, the precise molecular mechanisms and critical functional players of DNA demethylation in this process remain largely unexplored. Ten–eleven translocation (Tet) proteins convert 5-methylcytosine (5mC) to 5-hydroxymethylcytosine (5hmC) and its downstream derivatives. Here we show that 5hmC is elevated during the differentiation of adult neural stem cells (aNSCs), and Tet2 is primarily responsible for modulating 5hmC dynamics. Depletion of Tet2 leads to increased aNSC proliferation and reduced differentiation *in vitro* and *in vivo*. Genome-wide transcriptional analyses reveal important epigenetic roles of Tet2 in maintaining the transcriptome landscape related to neurogenesis. Mechanistically, transcription factor forkhead box O3 (Foxo3a) physically interacts with Tet2 and regulates the expression of genes related to aNSC proliferation. These data together establish an important role for the Tet2-Foxo3a axis in epigenetically regulating critical genes in aNSCs during adult neurogenesis.

[1] The Children's Hospital and Institute of Translational Medicine, School of Medicine, Zhejiang University, Hangzhou 310029, China. [2] Department of Human Genetics, Emory University School of Medicine, Atlanta, Georgia 30322, USA. [3] Department of Biostatistics and Bioinformatics, Rollins School of Public Health, Emory University, Atlanta, Georgia 30322, USA. [4] Guangdong-Hongkong-Macau Institute of CNS Regeneration, Joint International Research Laboratory of CNS Regeneration, Ministry of Education of PRC, Jinan University, Guangzhou 510632, PR China. [5] Sylvester Comprehensive Cancer Center, Department of Biochemistry and Molecular Biology, University of Miami Miller School of Medicine, Miami, Florida 33136, USA. [6] Department of Chemistry, University of Chicago, Chicago, Ilinois 60637, USA. [7] College of Business, Iowa State University, Ames, Iowa 50011, USA. * These authors contributed equally to this work. Correspondence and requests for materials should be addressed to X.L. (email: xuekun_li@zju.edu.cn) or to P.J. (email: peng.jin@emory.edu).

Epigenetic regulation represents a fundamental mechanism that maintains cell-type-specific gene expression during development and serves as an essential mediator to interface between the extrinsic environment and intrinsic genetic programs[1]. DNA methylation and histone modifications are the predominant forms of epigenetic regulations. Combined, these modifications in the context of chromatin have been described as 'epigenetic codes' that underpin gene expression programs and determine cell identity. At the DNA level, methylation of cytosine at the 5-carbon position (5-methylcytosine; 5mC) has been well studied as a critical epigenetic mark in the mammalian genome[2]. Cell-type-specific landscapes of methylation patterns, the methylome, confer one of the bases for cell-type-specific gene expression, resulting in vastly diverse cellular phenotypes and functions out of identical genetic materials[3]. Moreover, dynamic regulation of DNA methylation in response to various neuronal activities in the brain is critical for normal brain function[4,5].

5mC used to be viewed as a stable and long-lasting covalent modification to DNA; however, the fact that ten–eleven translocation (TET) proteins, including TET1, TET2 and TET3, could convert 5mC to 5-hydroxymethylcytosine (5hmC), giving us a new perspective on the previously observed plasticity in 5mC-dependent regulatory processes[6]. Furthermore, TET enzymes are also known to further oxidize 5hmC into 5-formylcytosine (5fC) and 5-carboxylcytosine (5caC), which can be readily repaired by DNA repair enzymes[7,8]. Present evidence indicates that 5hmC-mediated epigenetic mechanisms regulate gene transcription[9,10] and affect the differentiation of pluripotent stem cells[11,12].

Adult neurogenesis occurs in discrete regions of the adult mammalian brain, namely the subventricular zone (SVZ) of the lateral ventricles and the subgranular zone (SGZ) of the dentate gyrus (DG) in the hippocampus[13]. The newly generated neurons then integrate into the neuronal circuitry and modulate neural plasticity. Aberrant adult neurogenesis is linked to intellectual disabilities and neuropsychiatric and neurodegenerative disorders[14]. Emerging evidence suggests that various epigenetic mechanisms, including DNA methylation, histone modifications and non-coding RNAs, as well as cross-talk among these mechanisms, play important roles in fine-tuning and coordinating gene expression during adult neurogenesis[15]. TET-mediated oxidation of 5mC presents a particularly intriguing epigenetic regulatory paradigm in the mammalian brain, where its dynamic regulation is critical. In brain, 5hmC is significantly enriched relative to many other tissues and cell types[16–20]. 5hmC, but not 5mC, is found to increase during embryonic neurodevelopment and is associated with activated neuronal functional genes[21]. Genome-wide profiling during neurodevelopment and aging suggests that 5hmC is not only acquired during postnatal neurodevelopment and aging, but also displays spatial and temporal dynamics, pointing to its important role(s) in normal brain function[17]. Indeed, recent research has shown that Tet1 regulates activity-induced gene expression in neurons and modulates the formation and extinction of specific types of memory[4,22,23]. Tet1 also regulates neurogenesis by influencing DNA methylation states at the promoters of selective genes related to neurogenesis[24]. However, the precise epigenetic role(s) of 5hmC in adult neurogenesis, particularly in balancing adult NSC proliferation and differentiation plasticity, remains largely unexplored. Furthermore, the specific neuronal functions of individual Tet proteins and their functional partners during adult neurogenesis remain unclear.

Here we show that 5hmC abundance increases upon adult neural stem cell (aNSC) differentiation, accompanied by the specific elevation of Tet2 protein. We found that loss of Tet2 shifted the balance of aNSC states by promoting proliferation and impairing their differentiation ability *in vitro* and *in vivo*. Genome-wide transcriptional analyses echoed the phenotypic observations at the molecular level that the loss of Tet2 results in significant alteration of the transcriptome landscape related to aNSC proliferation and differentiation. Genome-wide 5hmC profiling suggested a positive correlation between 5hmC intragenic dynamics and a large portion of transcriptional upregulation. Mechanistically, we found that transcription factor forkhead box O3 (Foxo3a) interacted and coordinated with Tet2 to directly regulate proliferation-related genes. The binding of Foxo3a was significantly increased on the promoters of these genes in $Tet2^{-/-}$ aNSCs, concurring with the elevated 5hmC at these loci, to stimulate their transcription. *In vitro* analyses revealed that Foxo3a could preferentially bind to the motifs with 5hmC marks. Knockdown of Foxo3a in $Tet2^{-/-}$ aNSCs could rescue the deficits associated with the loss of the Tet2. These data together establish an important epigenetic role for the Tet2-Foxo3a axis in adult neurogenesis.

## Results

**Tet2 is important for 5hmC dynamics during adult neurogenesis.** Tet-mediated DNA demethylation is implicated in the regulation of gene expression during neurodevelopment and neurogenesis[21,25,26]. To dissect the specific and precise roles of Tet homologs, as well as cytosine oxidation dynamics during adult neurogenesis, we first isolated *in vivo* Nestin$^+$/Sox2$^+$ aNSCs from the dentate gyrus of 2-month-old wild-type (WT) adult mice (Supplementary Fig. 1a–d). These aNSCs could proliferate (referred as proliferating aNSCs or aNSCs hereafter) and produce both β-III tubulin (Tuj1)-positive neuronal cells and glial fibrillary acidic protein (GFAP)-positive glial cells after two days of differentiation[27,28] (referred as differentiated cells hereafter; Supplementary Fig. 1e–h), confirming their pluripotency and capacity for adult neurogenesis. There were few GFAP + /Nestin + double-positive cells in the differentiated conditions, indicating that the majority of cells have committed to either neuronal or glial lineage (Supplementary Fig. 1i–l). Co-immunofluorescence (IF) staining using 5hmC-specific antibody and the aNSC marker Nestin showed that 5hmC resided in the nuclei of Nestin-positive aNSCs (Fig. 1a–d). Interestingly, a significant and specific increase of Tet2 expression among all three Tet proteins was found in differentiated aNSCs (Fig. 1e). In the meantime, a significant increase of global 5hmC level was also observed (Fig. 1f,g, WT_P versus WT_D, $P < 0.01$, unpaired $t$-test). These data were in line with the previous observation that 5hmC accumulated during embryonic and postnatal neurodevelopment[17,21]. In contrast, differentiation from embryonic stem cells (ESCs) to embryonic bodies (EBs) led to a significant reduction in 5hmC (Fig. 1h,i, ES versus 5 days EB differentiation, D5, $P < 0.001$, unpaired $t$-test), suggesting a unique pattern of 5hmC in aNSC differentiation. The increase of 5hmC during aNSC differentiation was further confirmed using a highly sensitive mass spectrometry (liquid chromatography–mass spectrometry (LC-MS/MS)) approach (Supplementary Fig. 2a, WT_P versus WT_D, $P < 0.0001$, unpaired $t$-test). The expression dynamics of three Tet homologs in mice DG during postnatal neurodevelopment were also examined. Only Tet2 displayed a specific and significant increase during postnatal neurodevelopment compared to other Tet proteins (Supplementary Fig. 2b, $P < 0.01$, unpaired $t$-test), consistent with the observations in aNSCs (Fig. 1e). Tet1 showed significant downregulation, whereas Tet3 was marginally elevated (Supplementary Fig. 2b, $P < 0.01$ and $P = 0.35$, unpaired $t$-test). These data led us to further focus on Tet2 in adult neurogenesis.

To determine the essential roles of Tet2 in aNSC proliferation and differentiation, and its impact on adult neuronal functions,

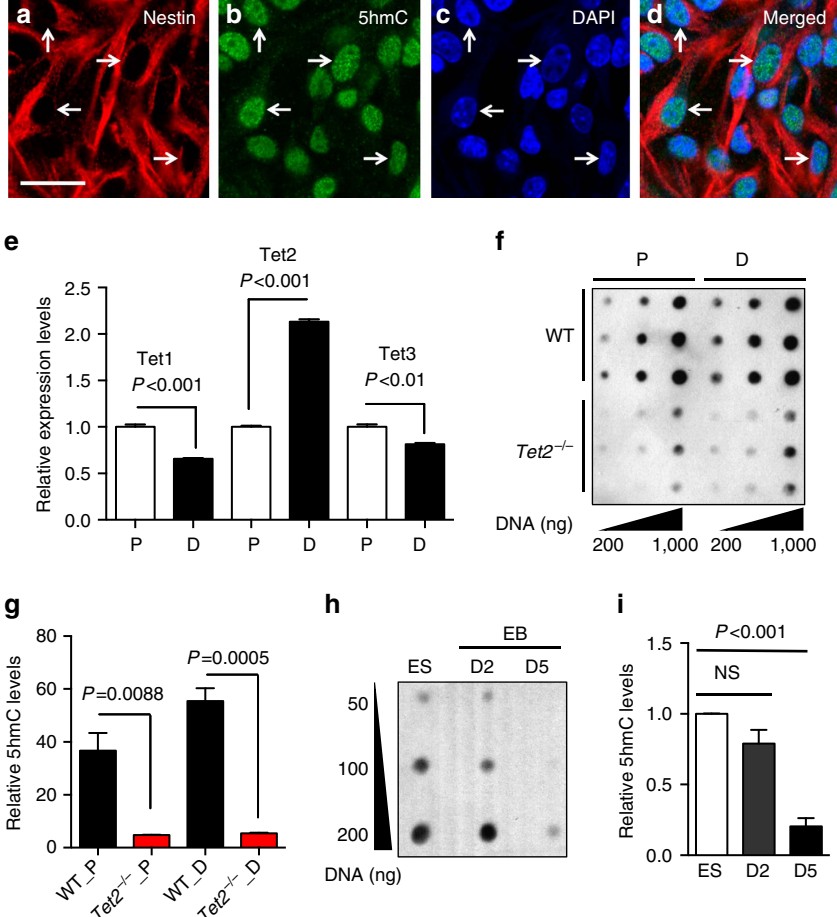

**Figure 1 | Tet2 is primarily responsible for 5hmC acquisition during the differentiation of adult neural stem cells.** (**a**–**d**) Representative immunostaining (IF) images of Nestin[+] adult neural stem cells (aNSCs) with (**a**) aNSC marker Nestin, (**b**) 5hmC, (**c**) nuclei DAPI staining and (**d**) merged images. 5hmC was primarily located in the nuclei of Nestin[+] aNSCs. Scale bar, 50 µm. $n = 3$. (**e**) Quantitative reverse-transcription PCR determined Tet1, Tet2, and Tet3 expression in proliferating aNSCs (P) and differentiated aNSCs (D). Tet2 displayed a specific elevation from P to D, whereas Tet1 and Tet3 were downregulated. All the data are presented as mean ± s.e.m. $n = 3$, unpaired $t$-test. $P$ values were indicated. (**f**) Representative dot-blot experiments using 5hmC-specific antibody indicated a relative increase of 5hmC levels upon aNSC differentiation, and Tet2 depletion resulted in a significant decrease of global 5hmC level in both aNSC-P and aNSC-D compared to WT cells. $n = 3$. (**g**) Quantification of 5hmC dot-blot intensities by ImageJ. $n = 3$, unpaired $t$-test. $P$ values were indicated. (**h,i**) Dot-blot assay and quantification indicated an overall reduction of 5hmC upon differentiation in mouse ES cells. $n = 3$, unpaired $t$-test. $P$ values were indicated.

we isolated *in vivo* Nestin[+]/Sox2[+] aNSCs from the brains of adult $Tet2^{-/-}$ mice by replacing part of exon 3 sequences of the $Tet2$ gene with nlacZ/nGFP as described previously[29], together with age-matched wild-type littermates. Tet2 depletion decreased global 5hmC levels in both proliferating and differentiated $Tet2^{-/-}$ aNSCs (Fig. 1f,g). Furthermore, loss of Tet2 consistently led to a significant decrease in 5hmC levels during postnatal brain development (Supplementary Fig. 2c,d). Thus, Tet2 is not only required to maintain 5hmC levels in proliferating aNSCs, it is also involved in the increased 5hmC levels upon aNSC differentiation. These data together indicated a predominant role of Tet2 in modulating 5hmC level during adult neurogenesis and neurodevelopment.

**Tet2 regulates adult neurogenesis *in vitro* and *in vivo*.** Given the roles of Tet2 in modulating 5hmC levels during neurogenesis, we examined whether loss of Tet2 could influence the proliferation and/or differentiation states of aNSCs. We first performed a BrdU (an analog of thymidine) incorporation assay to assess the effects of Tet2 ablation on aNSC proliferation. IF staining of BrdU showed more BrdU[+] cells in $Tet2^{-/-}$ aNSCs compared to WT (Fig. 2c,d comparing to Fig. 2a,b, respectively), and stereology quantification indicated a 51.7% increase of BrdU[+] cells in $Tet2^{-/-}$ aNSCs relative to WT cells (Fig. 2e, $P < 0.05$, unpaired $t$-test), suggesting that Tet2 depletion stimulated aNSC proliferation. The increased proliferation in $Tet2^{-/-}$ aNSCs *per se* was further supported by the BrdU/Nestin and BrdU/Sox2 co-staining (Supplementary Fig. 3a,b). We then surveyed the effect of Tet2 depletion on the differentiation of aNSCs. IF staining experiments showed that loss of Tet2 led to fewer neuronal marker Tuj1-positive neurons (Fig. 2j compared to Fig. 2f) and glial cell marker GFAP-positive glial cells (Fig. 2k compared to 2g). The quantification indicated 71.8% fewer Tuj1[+] neuronal cells (Fig. 2n, $P < 0.01$, unpaired $t$-test) and 45.1% fewer GFAP[+] glial cells (Fig. 2o, $P < 0.05$, unpaired $t$-test), indicating the loss of Tet2 could impair aNSC differentiation.

To validate the effects of Tet2 depletion on aNSC differentiation, we analyzed the promoter activity of pan-neuronal transcription factor NeuroD1 and glial cell marker GFAP in WT aNSCs using a previously described dual luciferase reporter assay[30]. We found that the loss of Tet2 could significantly

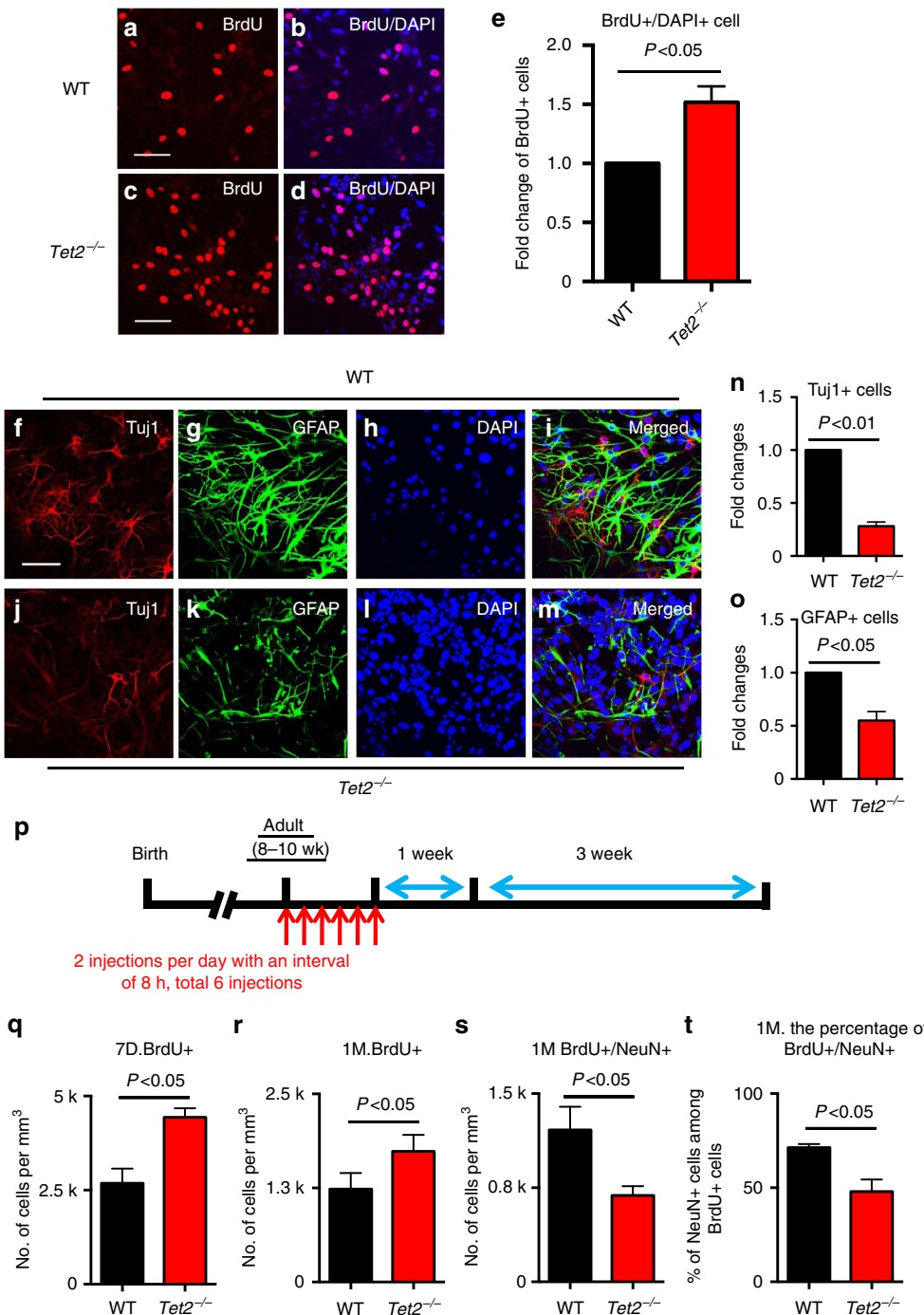

**Figure 2 | Tet2 regulates the balance of aNSC proliferation and differentiation both *in vitro* and *in vivo*.** (**a**–**d**) Representative BrdU staining in both WT (**a**) and *Tet2*<sup>−/−</sup> (**c**) aNSCs to access their proliferation states. Tet2 depletion promoted aNSC proliferation. (**b**,**d**) Overlay between BrdU and DAPI staining. Scale bar, 50 μm. $n = 3$. (**e**) Quantification of BrdU$^+$ aNSCs from WT and *Tet2*$^{-/-}$ mice in **a**–**d** indicated that depletion of Tet2 significantly promoted aNSC proliferation ($n = 3$; unpaired $t$-test, $P < 0.05$). (**f**–**m**) Representative IF staining in WT and *Tet2*$^{-/-}$ differentiated neural cells from isolated *in vivo* aNSCs with neuronal cell marker Tuj1 (**f**,**j**) and glial marker GFAP (**g**,**k**). The loss of Tet2 leads to a decrease of Tuj1$^+$ neurons and GFAP$^+$ glial cells. DAPI staining (**h**,**l**) and merged overlay (**i**,**m**) are indicated. Scale bar, 50 μm. $n = 4$. (**n**,**o**) Quantification of Tuj1$^+$ cells (**n**) and GFAP$^+$ glial cells (**o**) in WT and *Tet2*$^{-/-}$ neural cells differentiated from aNSCs ($n = 4$; unpaired $t$-test, $P < 0.01$ or 0.05). Fewer Tuj1$^+$ neuronal cells and GFAP$^+$ glial cells were found *Tet2*$^{-/-}$. (**p**) A diagram shows the experimental scheme to investigate adult neurogenesis in WT and *Tet2*$^{-/-}$ mice *in vivo*. WT and *Tet2*$^{-/-}$ mice were administered BrdU and killed at 1 or 4 weeks afterwards to assess the aNSC proliferation and differentiation states in the presence or absence of Tet2. (**q**) Quantification of BrdU$^+$ cells in the hippocampus of WT and *Tet2*$^{-/-}$ mice killed 7 days after final BrdU administration. Depletion of Tet2 resulted in a significant increase of aNSC proliferation (WT, $n = 4$; *Tet2*$^{-/-}$, $n = 3$; unpaired $t$-test, $P < 0.05$). (**r**) Quantification of BrdU$^+$ cells in the hippocampus of WT and *Tet2*$^{-/-}$ mice 4 weeks after the final BrdU administration. Depletion of Tet2 resulted in a significant increase of aNSC proliferation ($n = 5$; unpaired $t$-test, $P < 0.05$). (**s**) Quantification of BrdU$^+$/NeuN$^+$ cells in the hippocampus of WT and *Tet2*$^{-/-}$ mice 4 weeks after the final BrdU administration ($n = 4$; unpaired $t$-test, *$P < 0.05$). (**t**) Quantification of the percentage of BrdU$^+$/NeuN$^+$ cells among total BrdU$^+$ cells in the hippocampus of WT and *Tet2*$^{-/-}$ mice 4 weeks after the final BrdU administration ($n = 4$,; unpaired $t$-test, *$P < 0.05$).

decrease the promoter activities of NeuroD1 and GFAP (Supplementary Fig. 4a, $P < 0.0001$, unpaired $t$-test). In addition, the expression of the neural stem cell marker Nestin in $Tet2^{-/-}$ aNSCs is significantly higher than in WT (Supplementary Fig. 4b, $P < 0.01$, unpaired $t$-test). We then quantified the expression of several established neuronal and glial marker mRNA levels and found that the expression of neuronal cell markers β III-Tubulin (Tuj1) and NeuroD1 significantly decreased (55% and 94.4%, respectively) in $Tet2^{-/-}$ aNSCs relative to WT (Supplementary Fig. 4c,d, $P < 0.001$, unpaired $t$-test). The expression of glial cell markers GFAP and s100β also displayed significant reductions (95.7% and 58%, respectively) (Supplementary Fig. 4e,f, $P < 0.0001$, unpaired $t$-test). Moreover, siRNA knockdown of Tet2 in WT aNSCs resulted in significant downregulation of NeuroD1 and GFAP luciferase activities (Supplementary Fig. 4g,h, $P < 0.01$ and $P < 0.05$, unpaired $t$-test). To confirm the direct roles of Tet2 in modulating adult neurogenesis, we performed electroporation to re-express Tet2 in the $Tet2^{-/-}$ aNSCs. The re-expression of Tet2 significantly restored the NeuroD1 and GFAP reporter expression comparing to $Tet2^{-/-}$ aNSCs, supporting the key role of Tet2 in adult neurogenesis (Supplementary Fig. 4i,j).

To validate the important role of Tet2 in adult neurogenesis, we transiently knocked down Tet2 in wildtype aNSCs by shRNA electroporation (Supplementary Fig. 5a). The Tet2 knockdown led to the upregulation of Nestin and downregulation of NeuroD1 and GFAP, consistent with the increased proliferation and impaired differentiation in $Tet2^{-/-}$ aNSCs (Supplementary Fig. 5b). Importantly, simultaneous knockdown of both Tet1 and Tet2 in wildtype aNSCs resulted in a significant decrease of both NeuroD1 and GFAP reporter expression, indicating the double-knockdown impaired aNSC differentiation, the same effects observed in $Tet2^{-/-}$ and Tet2 shRNA knockdown aNSCs (Supplementary Fig. 5c,d). On the other hand, the depletion of both Tet1 and Tet2 significantly enhanced the aNSC proliferation rate, evidenced by significant increase of BrdU-positive cells (Supplementary Fig. 5e,f). These observations indicated the Tet1 and Tet2 double knockdown in adult NSCs displayed similar characteristics comparing to $Tet2^{-/-}$ aNSC. Collectively, our data strongly support the predominant roles of Tet2 in regulating adult neurogenesis, and loss of Tet2 shifted the balance of aNSC states by promoting proliferation and impairing their differentiation.

To further investigate the roles of Tet2 in regulating adult neurogenesis *in vivo*, both $Tet2^{-/-}$ and WT mice were administered BrdU and killed at 1 or 4 weeks afterwards (Fig. 2p). We first performed BrdU and Ki67 IF staining to assess the proliferation of aNSCs *in vivo* and found that most BrdU$^+$ cells and Ki67 + cells localized at the subgranular zone (SGZ) at the 1-week time point. Furthermore, there were significant higher BrdU$^+$ cells and Ki67 + cells in $Tet2^{-/-}$ mice (Fig. 2q, $P < 0.05$, unpaired $t$-test and Supplementary Fig. 6a). This observation was consistent with the notion that the SGZ is one of the main neurogenic regions *in vivo*[31,32], and confirmed that Tet2 knockout resulted in the increased aNSC proliferation *in vivo*. At the 4-week time point, some newborn neurons migrated to the granular zone and became mature neurons (Supplementary Fig. 6b, NeuN staining), and total BrdU$^+$ cells in $Tet2^{-/-}$ mice were still significantly higher than in WT mice (Fig. 2r; Supplementary Fig. 6b, $P < 0.05$, unpaired $t$-test). However, there were significantly fewer (43% fewer) newborn neurons (BrdU$^+$/NeuN$^+$) in $Tet2^{-/-}$ mice than in WT mice (Fig. 2s; Supplementary Fig. 6b, $P < 0.05$, unpaired $t$-test). In addition, the percentage of newborn neurons (BrdU$^+$ NeuN$^+$) from $Tet2^{-/-}$ mice was significantly lower than WT mice after 4 weeks (Fig. 2t; Supplementary Fig. 6b, $P < 0.05$,

unpaired $t$-test). No significant difference was found in hippocampus volume (Supplementary Fig. 6c, $P > 0.4$, unpaired $t$-test) and the survival rate of aNSCs between WT and $Tet2^{-/-}$ mice (Supplementary Fig. 6d, $P > 0.4$, unpaired $t$-test). Moreover, depletion of Tet2 also resulted in the significant increase of BrdU$^+$ cells in subventricular zone (SVZ), another brain regions harbouring aNSCs (Supplementary Fig. 7a,b, $P < 0.001$, unpaired $t$-test). These observations suggested the critical and general roles of Tet2 in regulating neurogenesis at different regions of the brain, and demonstrated Tet2 was pivotal to maintain proper aNSC-mediated neurogenesis both *in vitro* and *in vivo*.

**Tet2 regulates the expression of neurogenic genes**. To understand the molecular mechanism(s) underlying the adult neurogenesis associated with the loss of Tet2, we applied RNA-seq to profile the global transcriptome from $Tet2^{-/-}$ aNSCs and their WT controls to first assess how loss of Tet2 could influence genome-wide transcriptome in functional adult aNSCs. RNA-seq analyses indicated that Tet2 depletion resulted in 703 significantly upregulated and 979 downregulated genes (Fig. 3a; Supplementary Data 1, average $P$ value 0.0031 and 2.27E−03, respectively, cuffdiff $t$-test). Interestingly, gene ontology (GO) analysis revealed specific enrichment of upregulated genes in the cell cycle and DNA replication pathways. In contrast, the downregulated genes were largely enriched in GO terms related to neuronal functions, such as glutamate receptor signalling, synapse assembly, and dendrite morphogenesis (Fig. 3b; Supplementary Fig. 8a). We also performed RNA-seq in WT and $Tet2^{-/-}$ aNSC differentiated cells and compared the expression changes of these genes in differentiated cells. This panel of upregulated genes in the absence of Tet2 genes (Fig. 3b) appeared to be explicitly associated with aNSCs, as these genes did not show obvious differences in the differentiated WT or $Tet2^{-/-}$ cells (Fig. 3c, aNSC versus Differentiation). However, the patterns of downregulated genes showed a more similar trend between aNSC and differentiated cells than upregulated genes (Supplementary Fig. 8b), suggesting the upregulated genes are *bona fide* to the skewed aNSC characteristics in $Tet2^{-/-}$ aNSCs. Closer examination of these differentially expressed genes in protein interaction networks suggested that many of them are related to cell proliferation and neuronal differentiation, such as the mitotic sister chromatid segregation and glutamate receptor signalling pathways (Fig. 3d,e; Supplementary Fig. 8c,d). The expression of several downregulated genes or upregulated genes in $Tet2^{-/-}$ aNSCs involved in neuronal differentiation or cell growth/proliferation were further confirmed by qPCR (Supplementary Fig. 9a,b). Taken together, the global alteration of the transcriptome landscape in $Tet2^{-/-}$ aNSCs is in line with the phenotypic observations that Tet2 depletion led to enhanced proliferation and impaired neuronal differentiation.

**5hmC dynamics correlates with transcriptomic alterations**. Genome-wide 5hmC profiling from WT and $Tet2^{-/-}$ aNSCs were obtained using the previously described chemical-based 5hmC-containing DNA enrichment coupled with high-throughput sequencing (hMe-Seal)[18]. $Tet2^{-/-}$ aNSCs showed 84% fewer unique 5hmC peaks compared to WT, consistent with the dot blot results (Supplementary Fig. 10a). No significant difference of Tet1 or Tet3 expression was found when Tet2 was depleted (Supplementary Fig. 10b, unpaired $t$-test), consistent with the Tet2 is primarily responsible for adult neurogenesis. Nonetheless, Tet2 western blotting demonstrated the complete depletion of Tet2 at the protein level (Supplementary Fig.10c).

Recent studies suggested a positive correlation between intragenic 5hmC with transcription control during neuronal

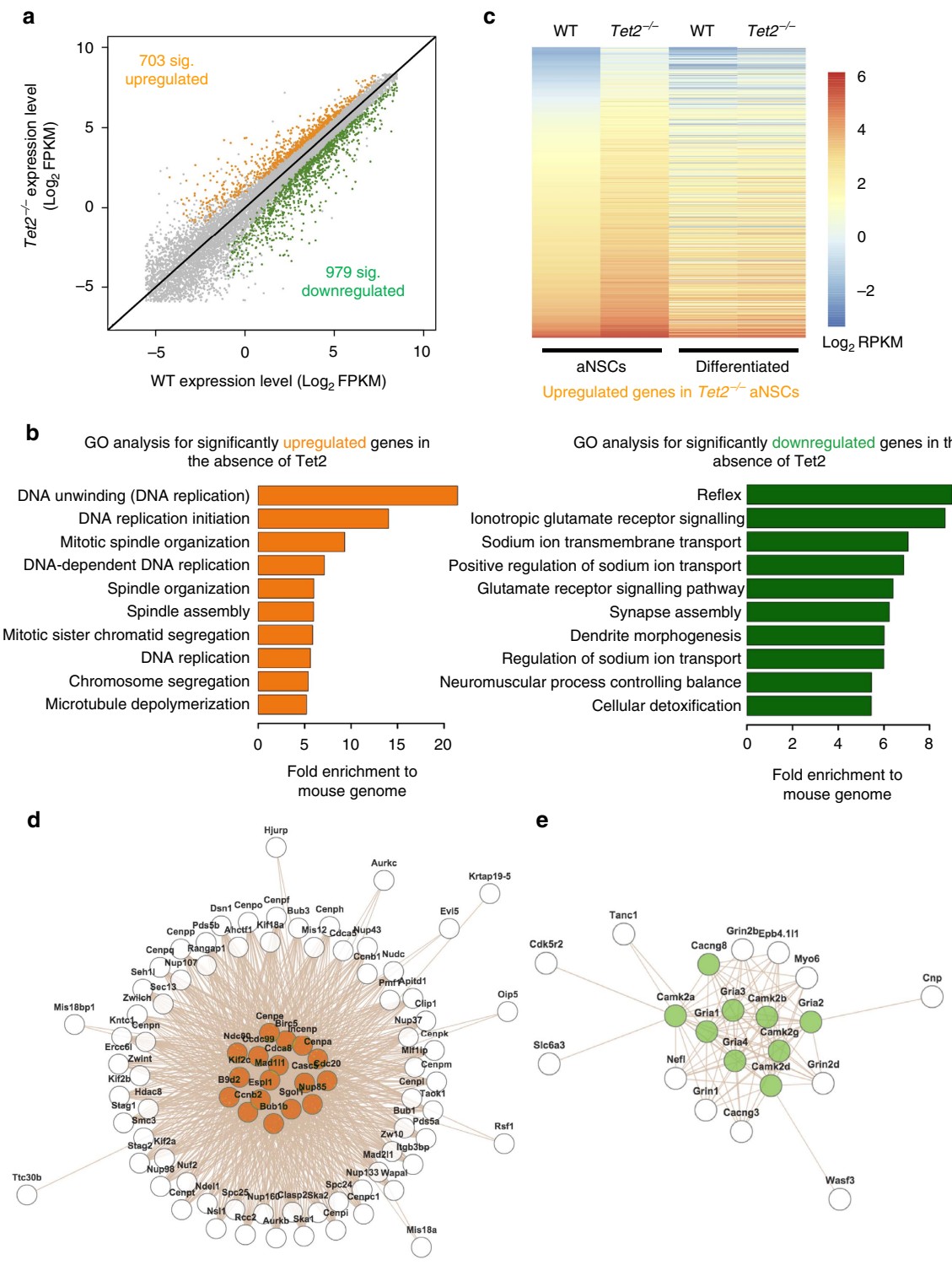

**Figure 3 | Loss of Tet2 led to a significant alteration of global transcriptome related to adult neurogenesis.** (**a**) RNA-seq experiments were performed to evaluate the global gene expression change $Tet2^{-/-}$ versus wildtype aNSCs ($n = 2$). Logarithmic scale (Log) of Fragments Per Kilobase Of Exon Per Million Fragments Mapped (FPKM) values were plotted, with significantly up- and downregulated genes ($P < 0.05$) highlighted in orange and green, respectively ($P < 0.05$). (**b**) Gene ontology (GO) analysis was performed using significantly up- and downregulated genes in **a**. Most significant GO pathways (fold enrichment) were indicated in the bar graph. Significantly upregulated genes were enriched in GO terms involved in the cell cycle and DNA replication pathways, whereas downregulated genes participated in GO terms related to neuronal functions. (**c**) Log scale of FPKM values of 703 significantly upregulated genes from WT and $Tet2^{-/-}$ aNSCs and differentiated cells were demonstrated by heatmap view. These genes displayed aNSC-specific dynamics without significant changes in Differentiated cells. (**d,e**) Significantly up- and downregulated genes in the absence of Tet2 were further subjected to protein interaction network module analysis by the Web-based Gene Set Analysis Toolkit (WebGestalt)[58]. Networks of several genes involved in either the mitotic sister chromatid segregation or glutamate receptor signalling pathway are shown.

development[21,33]. We sought to investigate whether Tet2 was primarily involved in controlling intragenic 5hmC dynamics and thus epigenetically modulating aNSC signature transcripts during adult neurogenesis. Since 5hmC has also been implicated in distal regulatory regions[18,34] that can be specifically controlled by Tet2 (ref. 35), we investigated the 5hmC dynamics between 5-kb upstream of transcription start sites (TSS) and downstream of transcription end sites (TES). Metagene plots of normalized 5hmC reads on 703 significantly upregulated genes and 979 downregulated genes revealed an overall positive correlation between 5hmC and transcription in WT and $Tet2^{-/-}$ aNSCs. On upregulated genes, average 5hmC showed a general increase in $Tet2^{-/-}$ aNSCs, including promoters (1 kb $\pm$ TSS) and gene bodies (Fig. 4a). On the other hand, 5hmC was primarily reduced between promoters to the first two thirds of gene bodies (66% average position normalized to gene length), but showed a modest increase on promoters and the last third of the gene bodies (Fig. 4b).

Since the 703 significantly upregulated genes emerged to be specific between WT and $Tet2^{-/-}$ aNSCs (Fig. 3c), they were likely to reflect the primary phenotypic effects in $Tet2^{-/-}$ aNSCs. In addition, Tet2 knockdown is known to directly increase intragenic 5hmC in ES cells[36], in line with our observations in aNSCs (Fig. 4a). We thus focused on understanding the molecular mechanisms of Tet2 and 5hmC in modulating these upregulated genes. We first calculated the average intragenic 5hmC levels on each individual upregulated gene and found 62% (439, $P < 0.0001$, binomial test) of the 703 upregulated genes bore higher 5hmC tag density in $Tet2^{-/-}$ than WT (Fig. 4c). GO analyses on the 439 genes with higher intragenic 5hmC in $Tet2^{-/-}$ aNSCs preserved the key pathways related to DNA replication and mitosis (Fig. 4d). In comparison, GO analyses on the 38% of upregulated genes with reduced 5hmC in $Tet2^{-/-}$ were largely involved in protein depolymerization and disassembly (Supplementary Fig. 10d). The sequencing results of the hME-Seal were further validated by loci-specific

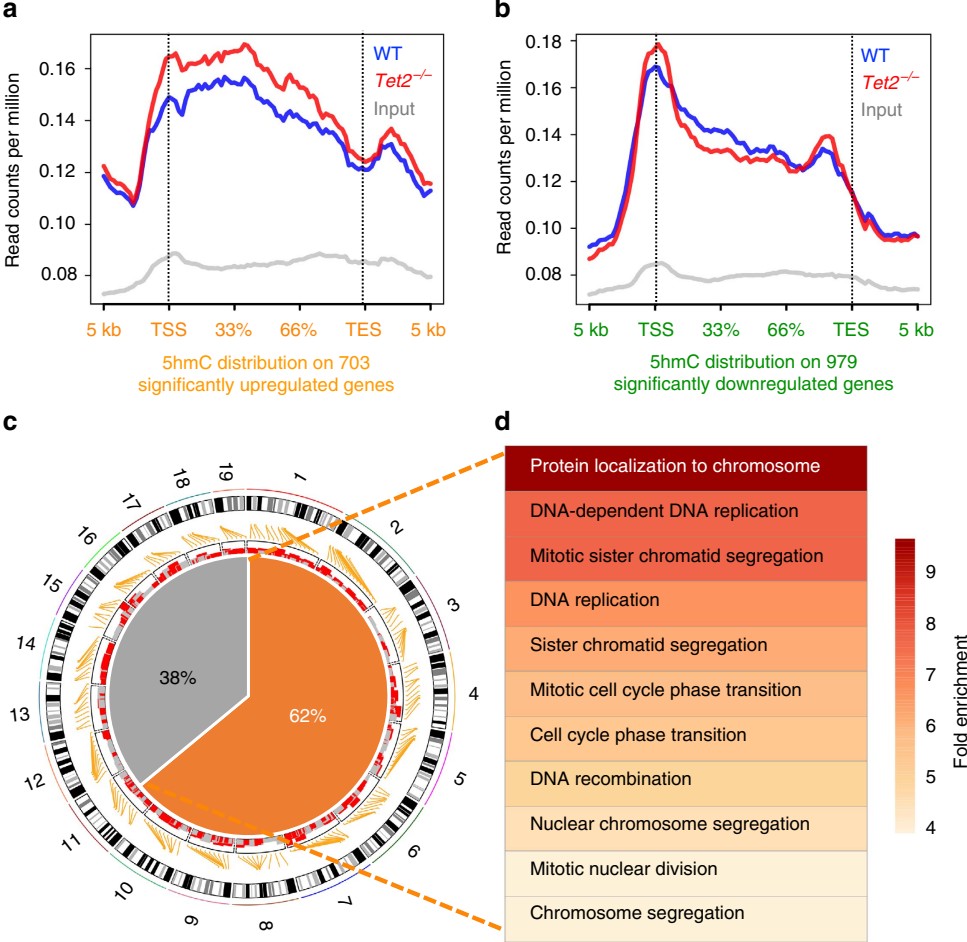

**Figure 4 | Intragenic 5hmC dynamics largely correlated with transcriptome alterations in aNSCs.** (**a,b**) Significantly upregulated (**a**, $n = 703$) and downregulated genes (**b**, $n = 979$) in aNSCs in the absence of Tet2 were normalized to 1–100% as relative positions. 5 kb upstream of transcription start sites (TSS) and downstream of transcription end sites (TES) were included. Average normalized 5hmC mapped reads per million from WT, $Tet2^{-/-}$, and non-enriched input DNA were calculated in each bin of each gene, and average bin values from all genes were plotted. 5hmC in $Tet2^{-/-}$ showed higher enrichment on the gene bodies of upregulated genes and lower enrichment on downregulated genes compared to 5hmC from WT aNSCs. The entire transcripts from transcription start sites (TSS) to transcription ending sites (TES) were referred as intragenic regions. (**c**) Circular map view of upregulated genes in the absence of Tet2 and their correlation with 5hmC are indicated. Each mouse chromosome is shown in the black-and-white outer track. Orange lines attached to the chromosomes indicate the chromosomal locations of 703 upregulated genes. Intragenic 5hmC reads ratio between WT and $Tet2^{-/-}$ aNSCs were calculated on these genes and indicated in the inner track, with red bars demonstrating higher intragenic 5hmC in $Tet2^{-/-}$ aNSCs and grey bars showing lower intragenic 5hmC in $Tet2^{-/-}$ aNSCs. Pie chart summarizes these results and indicates 62% of upregulated genes ($n = 439$; binomial test, $P < 0.0001$,) carried higher intragenic 5hmC in $Tet2^{-/-}$ aNSCs than WT. (**d**) GO analyses on 439 upregulated genes bearing higher intragenic 5hmC in $Tet2^{-/-}$ preserved GO terms related to the cell cycle and proliferation. Fold enrichment is demonstrated by heatmap view.

qPCR (Supplementary Fig. 11a,b). These observations indicated Tet2 and 5hmC directly regulated genuine genes correlated with aNSC proliferation, whose ectopic upregulation in the $Tet2^{-/-}$ aNSCs were corresponding to the balance shift between proliferation and differentiation.

**Foxo3a interacts with Tet2 to regulate adult neurogenesis**. Previously, along with others, we found that 5hmC is highly enriched within distal regulatory regions, particularly enhancer elements in embryonic stem cells[34,35,37]. To identify the potential transcription factor(s) that could be involved in Tet2-mediated epigenetic modulation in neurogenesis, we examined the published ChIP-seq datasets of multiple transcription factors that have been linked to neurogenesis previously, and overlapped them with the 5hmC dynamic regions regulated by Tet2. We observed that Foxo3a binding sites have the most overlap. Foxo3a, a mammalian forkhead family member, is well known to regulate gene expression and help preserve an intact pool of neural stem cells, at least in part by negatively regulating neuronal differentiation[38]. Analyses of genome-wide Foxo3a binding patterns in aNSCs revealed that Foxo3a is enriched at the enhancers of genes involved in neurogenic pathways[39]. Intriguingly, it has also been suggested Foxo3a could serve as a transcriptional activator[40]; however, detailed mechanistic insight into Foxo3a as a transcription factor moderating adult neurogenesis remains as an open question. To systematically investigate the regulatory role of Foxo3a in adult neurogenesis, especially its coordination with Tet2 and cytosine modifications, we performed our own Foxo3a chromatin immunoprecipitation coupled with high-throughput sequencing (ChIP-seq) in control and $Tet2^{-/-}$ aNSCs. 56.4% of our Foxo3a ChIP-seq peaks overlapped with published Foxo3a binding sites using different NSCs[39] (Supplementary Fig. 11c). It is now widely accepted that Tet proteins can influence proximal DNA modifications[36], thus we first separately merged the Foxo3a or 5hmC peaks from WT and $Tet2^{-/-}$ aNSCs (Supplementary Data 2) and tested their correlation by overlapping peaks within 500 bp of 5hmC peaks (Supplementary Data 3). Interestingly, 8,179 out of 21,395 Foxo3a peaks resided within a 500-bp range of 5hmC peaks (Fig. 5a, $P<0.0001$, binomial test), implying the potential orchestration between Foxo3a and Tet2 in aNSCs. To test this hypothesis, we asked whether Foxo3a is biochemically associated with Tet2 protein. We performed co-immunoprecipitation (co-IP) experiments followed by western blot analyses using WT aNSCs with anti-Foxo3a and anti-Tet2 antibodies, and found that endogenous Tet2 could be readily precipitated by Foxo3a and vice versa (Fig. 5b). To further confirm this observation, we performed additional co-IP experiments using HEK293 cells transfected with Myc-tagged full-length mouse Tet2 and FLAG-tagged mouse Foxo3a. The Tet2-Foxo3a binding was validated by reciprocal IP (Fig. 5c). We reasoned that, as a transcriptional activator, Foxo3a ought to be dynamically enriched in promoter regions and positively correlated with upregulated genes in the absence of Tet2. To test this, we focused on the Foxo3a dynamic changes on promoter regions of 439 upregulated genes modulated by Tet2 (Fig. 4c,d). Remarkably, 310 out of 439 (70.6%) of these genes displayed increased Foxo3a promoter binding in $Tet2^{-/-}$ aNSCs compared to WT (Fig. 5d, $P<0.0001$, binomial test), indicating these genes were likely under the control of both Tet2 and Foxo3a. GO analyses on these genes revealed similar enrichment of proliferation-related pathways, including DNA replication chromosome organization and the cell cycle. Interestingly, Foxo3a-regulated genes were also enriched in nervous system development, reflecting the neuronal signature of Foxo3a, besides its general roles in modulating aNSC proliferation (Fig. 5e).

We generated heatmaps using Foxo3a and 5hmC $\log_2$ normalized reads per million on promoters (500 bp upstream and downstream of transcription start sites (TSS)) and gene bodies (from 500 bp downstream of TSS to transcription end sites (TES)) of 310 genes (Fig. 5d) co-regulated by Tet2 and Foxo3a (Supplementary Fig. 12). Violin plots to quantify the mean of each heatmap lane with statistical analyses were also engaged. As shown in Supplementary Fig. 12, Foxo3a showed higher enrichment on promoters than gene bodies, and Tet2 depletion led to significantly increased Foxo3a binding to these promoters ($P<0.0001$, unpaired $t$-test). 5hmC levels were also significantly increased in the absence of Tet2 ($P<0.0001$, unpaired $t$-test). Both Foxo3a and 5hmC on gene bodies showed a modestly significant increase in $Tet2^{-/-}$ versus WT comparing to their bindings on promoters ($P<0.01$, unpaired $t$-test). These data present a mechanistic correlation of Foxo3a and 5hmC on promoters of these upregulated genes in the absence of Tet2. We chose two cohorts of upregulated genes co-regulated by Tet2 and Foxo3a from Supplementary Fig. 7 that are involved in the cell cycle ($n=38$) and nervous system development ($n=54$) to investigate the Foxo3a and 5hmC dynamics (Supplementary Data 4). Foxo3a showed the significant elevation on promoters of genes in both groups, again correlated with promoter 5hmC dynamics (Fig. 6a,b, $P=0.003$ and $P=0.01$, unpaired $t$-test).

To validate the predominant roles of Foxo3a in the regulation of the upregulated genes in $Tet2^{-/-}$ aNSCs, we examined the Foxo3a and 5hmC dynamic changes on the promoters and gene bodies among significantly upregulated and downregulated genes (Supplementary Fig. 13a). Both Foxo3a and 5hmC showed substantially more and significant enrichment on the promoters of upregulated genes than downregulated genes in $Tet2^{-/-}$ aNSCs comparing to WT (Supplementary Fig. 13a). These observations support the predominant roles of Foxo3a on upregulated genes by binding to their promoters. Overall Foxo3a and 5hmC changes on the gene bodies positively correlated with their expression, but the Foxo3a changes on the gene bodies of downregulated genes were not statistically significant (Supplementary Fig. 13b, $P=0.2$, unpaired $t$-test). These data suggested the positive correlation of Foxo3a and 5hmC primarily for promoter binding, and thus affect transcription involved in proliferation. In addition, these analyses further indicated that Foxo3a might not be the direct regulator for these downregulated genes.

Since specific and significant increase of Foxo3a binding was observed at the promoters of upregulated genes in $Tet2^{-/-}$ aNSCs and correlated with 5hmC accumulations *in vivo* (Supplementary Figs 12 and 13a), we sought to test whether 5hmC accumulation could directly recruit Foxo3a as a mechanism to stimulate these gene expression. To test this possibility, we synthesized two sets of biotin-labelled DNA oligos with identical sequences based on Foxo3a ChIP-seq top motifs. The cytosines in the consensuses sequence were either hydroxymethylated (5hmC probe) or unmodified (control probe), with the latter used as a negative control. Control and 5hmC-containing probes were used for *in vitro* binding assay using equal amount of Foxo3a recombinant protein. We found that Foxo3a displayed significantly stronger binding to the 5hmC-modified probes over control *in vitro* (Supplementary Fig. 14a,b). These data, together with Foxo3a/5hmC correlation *in vivo*, revealed the molecular mechanism of upregulated genes that 5hmC accumulation upon Tet2 deletion served as a binding platform to further recruit Foxo3a and stimulate gene expression.

To confirm the direct role of Foxo3a in regulating gene expression related to aNSC proliferation, several genes from Fig. 6a,b were examined by qPCR in WT, $Tet2^{-/-}$ aNSC and $Tet2^{-/-}$ aNSC with the knockdown of Foxo3a mRNA

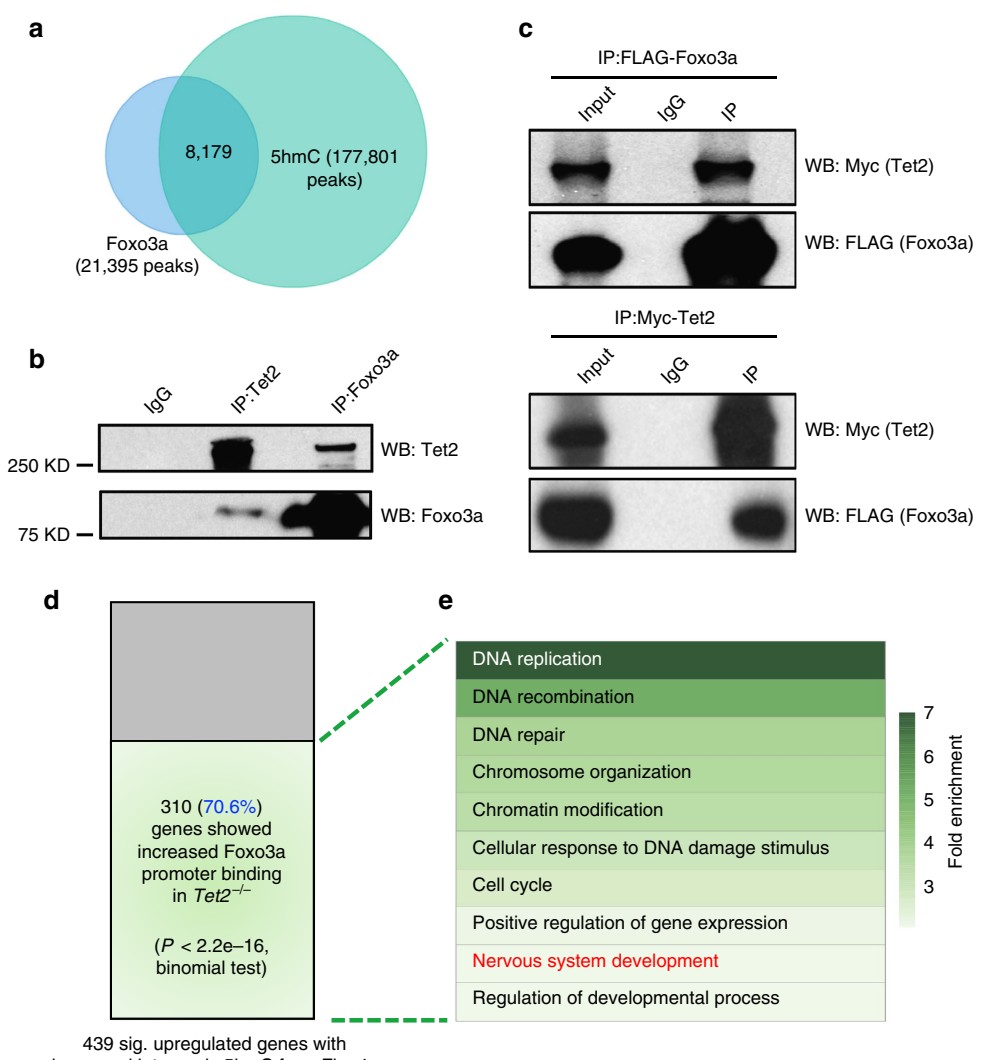

**Figure 5 | Foxo3a interacts with Tet2 and modulates the expression of key genes involved in neurogenesis.** (**a**) Merged Foxo3a ChIP-seq and 5hmC-seq peaks from WT and $Tet2^{-/-}$ aNSCs were used to calculate their overlap. 8,179 out of a total 21 395 Foxo3a peaks located 500 bp adjacent to aNSC 5hmC peaks (binomial test, $P < 0.0001$). (**b**) Endogenous Foxo3a and Tet2 co-immunoprecipitation (co-IP) experiments were performed using WT aNSCs, followed by immunoblotting with either Tet2 or Foxo3a antibody. Endogenous Foxo3a interacted with Tet2 in WT aNSCs. (**c**) Mouse Myc-Tet2 and FLAG-Foxo3a expression plasmids were co-transfected into HEK293 cells, and co-IP experiments were performed by anti-FLAG and anti-Myc antibodies followed by immunoblotting. Exogenous mouse Foxo3a and Tet2 interacted in reciprocal IP. (**d**) Average normalized Foxo3a mapped reads per million from WT and $Tet2^{-/-}$ were calculated on promoters (1 kb ± TSS) of 439 upregulated genes bearing higher intragenic 5hmC in $Tet2^{-/-}$ (Fig. 4d). Foxo3a became further enriched on 310 out of 439 (binomial test, $P < 0.0001$) genes in the $Tet2^{-/-}$ aNSCs. (**e**) GO analyses on 310 upregulated genes bearing higher intragenic 5hmC and promoter Foxo3a in $Tet2^{-/-}$ were genuine in GO terms related to the cell cycle and nervous system development. Fold enrichment is demonstrated by heatmap view. Selected terms with fold enrichment > 2 are indicated.

(Supplementary Fig. 14c). The Foxo3a knockdown significantly reduced the upregulation of these genes caused by the Tet2 depletion (Fig. 6c–e, unpaired *t*-test). Furthermore, Foxo3a knockdown significantly ameliorate the over-proliferation deficits in $Tet2^{-/-}$ aNSC (Fig. 6f), supporting an important role of Tet2-Foxo3a axis in the regulation of adult neurogenesis.

## Discussion

Our data presented here highlight the unique function of Tet2 in dynamic 5hmC regulation during adult neurogenesis. The loss of Tet2 significantly decreased 5hmC levels and partially blocked the acquisition of 5hmC upon aNSC differentiation. We found that the loss of Tet2 promoted aNSC proliferation but inhibited their differentiation both *in vitro* and *in vivo*. Global transcriptome analyses established the molecular bases of these phenotypic

observations, with significantly upregulated or downregulated genes involved in aNSC proliferation or neuronal differentiation, respectively. Mechanistically, we found that Tet2 interacted with the neuronal transcription activator Foxo3a and co-regulated key genes involved in aNSC proliferation. Tet2 depletion led to an ectopic increase of Foxo3a and 5hmC on promoters and intragenic regions of genes involved in the cell cycle and nervous system development. The preferential binding of Foxo3a to hydroxymethylated DNA were further confirmed *in vitro*. Thus, our data suggest a model wherein, in wild-type aNSCs, Tet2 and Foxo3a bind and co-regulate genes involved in normal aNSC proliferation and aNSC signature/plasticity maintenance. The presence of Tet2 on these genes maintains promoter and intragenic 5hmC homeostasis, which could determine the Foxo3a binding dynamics to keep its amount optimized on these regions (Fig. 7a). In the absence of Tet2, 5hmC accumulates on promoters

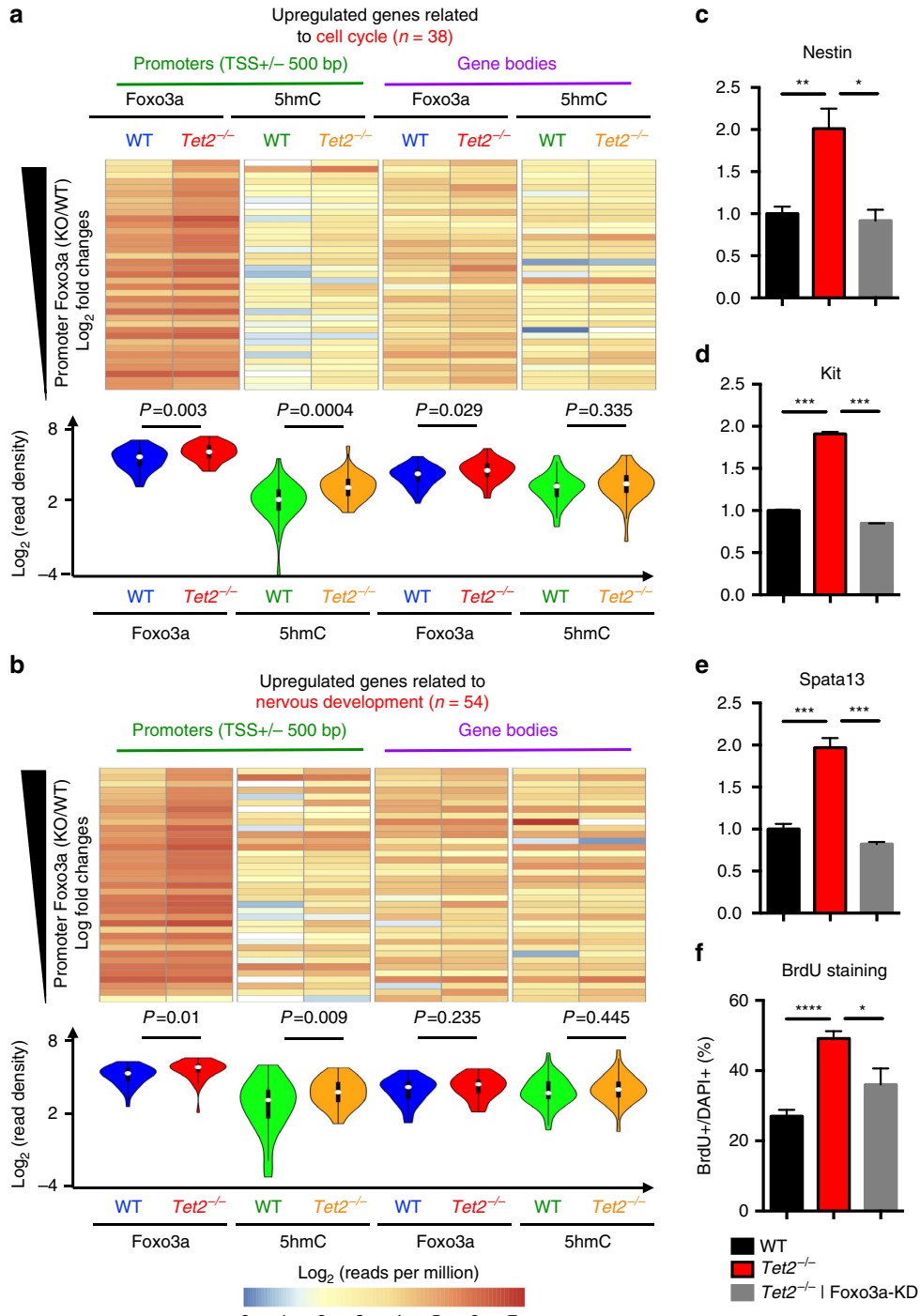

**Figure 6 | Foxo3a directly regulated genes involved in aNSC proliferation and signature by binding to their promoters.** (**a,b**) Average normalized Foxo3a and 5hmC mapped reads from both WT and $Tet2^{-/-}$ aNSCs were calculated on promoters and gene bodies of the cell cycle (**a**, $n = 38$) and nervous system development (**b**, $n = 54$) related genes. $Log_2$ of average reads per million on each gene were displayed by heatmap. Violin plots summarized the mean value of each column, and statistical significance was assessed by unpaired $t$-test. Foxo3a was substantially enriched on promoters and became significantly increased in the absence of Tet2. Genes involved in nervous system development showed general higher Foxo3a and 5hmC enrichment than genes involved in the cell cycle. $P$ values were indicated. Genomic regions between 500 bp upstream and downstream of transcription start sites (TSS) were referred as promoter regions. Genomic regions between 500 bp downstream of TSS and transcription end sites (TES) were referred as gene bodies. (**c–e**) Real-time PCR to determine the expression dynamics of selected loci related to the cell cycle and nervous system development. The expression of these genes was upregulated in the absence of Tet2 in aNSCs and restored when Foxo3a was knocked down by siRNA in $Tet2^{-/-}$ aNSCs. $n = 3$; unpaired $t$-test. *$P < 0.05$; **$P < 0.01$; ***$P < 0.001$. (**f**) Quantification of BrdU$^+$ aNSCs from WT, $Tet2^{-/-}$ and Foxo3a knockdown in $Tet2^{-/-}$ background. The relative percentage of BrdU$^+$ aNSCs were normalized to total cells by DAPI staining. $Tet2^{-/-}$ aNSCs displayed significant higher BrdU$^+$ aNSCs, indicating their higher proliferation rates. Depletion of Foxo3a in $Tet2^{-/-}$ aNSCs significantly reduced their proliferation. ($n = 3$; unpaired $t$-test, *$P < 0.05$; ***$P < 0.001$).

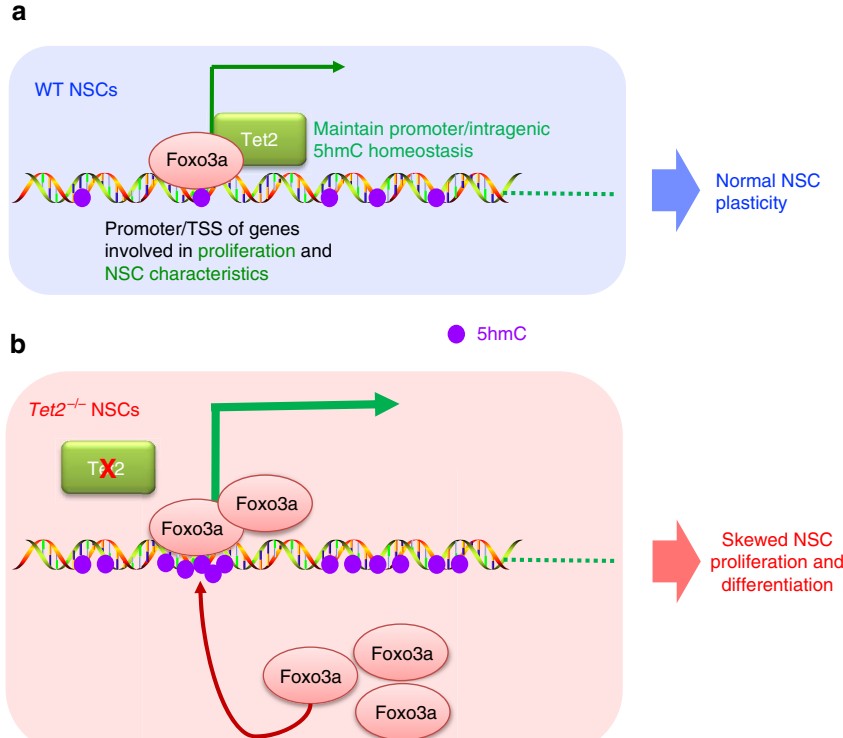

**Figure 7 | The Tet2-Foxo3a axis was critical in modulating aNSC transcriptome landscape involved in proliferation and aNSC characteristics.**
(**a**) In WT aNSCs, Tet2 and Foxo3a bind and co-regulate the genes involved in aNSC proliferation and aNSC signature/plasticity maintenance. The binding of Tet2 to these genes maintains promoter and intragenic 5hmC homeostasis, which could determine the Foxo3a binding dynamics to keep its amount optimized on these regions. (**b**) In the absence of Tet2, 5hmC accumulates on promoters and gene bodies due to the impaired DNA demethylation process and recruits additional Foxo3a to ectopically increase the expression of these genes and cause skewed aNSC proliferation and differentiation balance.

and gene bodies due to the impaired DNA demethylation process and recruits additional Foxo3a to ectopically increase the expression of these genes and cause a skewed aNSC proliferation and differentiation balance (Fig. 7b).

Although all three Tet homologues in mammals possess the indistinguishable ability to oxidize 5mC to 5hmC *in vitro*[11], due to their conserved catalytic domain in C-terminals, their protein structures, expression in various tissues and at different developmental stages, as well as their ability to control 5mC/5hmC oxidation loci genome-wide and influence gene expression, appear to be quite different[41,42]. Earlier work showed that loss of Tet1 leads to a decrease in the self-renewal capability of aNSCs, whereas its loss did not impact the differentiation potential[24]. This is the opposite of what we found here in $Tet2^{-/-}$ mice, indicating that Tet1 and Tet2 have distinct roles in the regulation of neurogenesis. Indeed, Tet1 was found to regulate neurogenesis by influencing DNA methylation states at the promoters of selective genes related to neurogenesis, while our study here demonstrated the important roles of a Tet2 interactor, Foxo3a, in targeting the promoters of genes related to aNSC proliferation. Nonetheless, simultaneous knockdown of Tet1 and Tet2 in aNSCs recapitulated the enhanced proliferation and impaired differentiation in $Tet2^{-/-}$ or Tet2 shRNA knockdown, suggesting a primary role of Tet2 in regulating adult neurogenesis.

Our molecular analyses suggested Tet2 epigenetically controlled a subset of genes related to proliferation and differentiation, and determined their levels during the transition. Depletion of Tet2 resulted in the global alteration of gene expression and 5hmC level. These molecular data fitted well to explain the phenotype observed in $Tet2^{-/-}$ aNSCs. A recent study in mouse ES cells revealed that Tet1 depletion decreases 5hmC in both TSS

and gene bodies, while Tet2 depletion impairs 5hmC levels primarily in gene bodies with a gain of 5hmC in TSS[36]. Our 5hmC profiling data in aNSCs showed the 5hmC on gene bodies seemed to correlate well with significant altered transcription. This positive correlation of 5hmC with gene expression appeared to be more specific in aNSCs than mouse ESCs since no correlation of 5hmC and gene expression was found in mESCs[36]. Interestingly a recent study showed depletion of Tet2 in mast cells can also promote mast cell proliferation and inhibit their differentiation into myeloid cells. Importantly, their data, in agreement with ours, suggested the control of mast cell proliferation appeared to be predominantly Tet2-dependent[27]. These facts together indicated a universal and specific role of Tet2 to prevent overgrowth of different cell types. It has been well established that various mutations of Tet2 lead to the development of myeloid malignancy[28,43,44], thus the Tet2 loss-of-function mutations or deletions in aNSCs could link aberrant neurogenesis to the development of brain tumours such as gliomas[45]. Furthermore, these results validated our approaches to focus on the aNSC upregulated genes in the absence of Tet2, such as oncogene Kit (Fig. 6d) that also appeared to be Tet2 target in mast cells[27].

Intense efforts have gone into pinpointing the detailed molecular mechanisms of how Tet proteins regulate gene expression through cytosine oxidation. Our present data revealed a novel Tet2 binding protein, Foxo3a, and a coordinative role of Tet2-Foxo3a in modulating aNSC proliferation and signature genes. It has been reported that Foxo3a is required to regulate the Notch signalling pathway during the self-renewal of adult muscle stem cells[46], and a recent publication suggests its role in defining enhancers in neurogenic genes[39]. Our data helped explain the molecular basis of differential gene expression regulated by

Foxo3a in the absence of Tet2. Although overall 5hmC levels are significantly reduced in the absence of Tet2, its levels and intragenic distributions on critical upregulated genes related to aNSC pluripotency and signature are actually elevated. These observations provided an alternative insight that Tet2 can not only increase 5hmC levels by converting 5mC to de novo 5hmC, but can also further modulate the cytosine oxidation dynamic by converting 5hmC to 5fC/5caC. This characteristic of Tet2 was particularly important to maintain the 5hmC homeostasis on critical genes and keep the amount of transcription factor binding, such as Foxo3a, in check. Thus, our findings here define a novel mechanism of the Tet2-Foxo3a axis in epigenetically modulating transcriptome dynamics involved in aNSC proliferation and characteristics.

## Methods

**Mice and isolation and culture of neural stem cells.** Mice were housed in the standard conditions of the Division of Animal Resources at Emory University on a 12-h light (07:00 to 19:00 hours)/dark (19:00 to 07:00 hours) cycle with free access to food and water. All animal procedures were performed according to protocols approved by the Emory University Animal Care and Use Committee and the University Committee on Animal Care of Soochow University and conducted in accordance with the guidelines on Animal Use and Care of the National Institutes of Health (NIH) and the ARRIVE (Animal Research: Reporting In Vivo Experiments). The generation of Tet2 knockout ($Tet2^{-/-}$) mice has been described previously[29]. The littermates of adult wild-type (WT) and $Tet2^{-/-}$ male mice were used in this study. Neural stem cells (NSCs) were isolated from the dentate gyrus of adult WT and $Tet2^{-/-}$ age-matched littermate mice (2 months old) and cultured as described previously[47].

For histological studies, mice were killed by intraperitoneal injection of chloral hydrate. Mice were then transcardially perfused with PBS followed by 4% PFA (prepared with PBS). Brains were dissected out, post-fixed with 4% PFA, and then equilibrated in 30% sucrose at 4 °C. Forty-micrometre floating brain sections were generated with a Leica microtome.

**In vitro and in vivo neurogenesis assays.** For the in vitro proliferation assay, aNSCs growing on coverslips were cultured with DMEM/F-12 medium containing 5 μM BrdU for 3 h. For the in vitro differentiation assay, aNSCs growing on coverslips were cultured with DMEM/F-12 medium containing 5 μM forskolin (Sigma) and 1 μM retinoic acid (Sigma) for 2 days. The aNSCs were fixed with 4% paraformaldehyde, and immunofluorescence was performed as described below.

For the in vivo neurogenesis assay, adult WT and $Tet2^{-/-}$ mice received two BrdU injections per day (50 mg kg$^{-1}$, i.p.) at an interval of 8 h for a total of six injections. To study proliferation, 1 week after the final BrdU administration, one group of mice was anesthetized with chloral hydrate (50 mg kg$^{-1}$) and perfused with PBS, followed by perfusion with 4% paraformaldehyde. To study differentiation, 4 wk after the final BrdU administration, another group of mice was killed and brains were removed for dehydration in 30% sucrose overnight. The 40-μm brain sections were collected with a cryostat (Leica), and immunofluorescence staining was performed. The following primary antibodies were used: mouse anti-rat Nestin (1:500 in IF, BD, Cat. 556309), rabbit anti-Sox2 (1:500 in IF, Millipore, Cat. AB5603), mouse anti-β-III-tubulin (1:2000 in IF, Promega, Cat. G7121), rabbit anti-glial fibrillary acidic protein (1:1000 in IF, DAKO, Cat. Z0334), rat anti-BrdU (1:2000 in IF, Abcam, Cat. ab1893), rabbit Ki-67 (1:200 in IF, Millipore, Cat. AB9260), and mouse anti-neuronal nuclei (1:500 in IF, Millipore, Cat. MAB377). The following secondary antibodies were used: goat anti-mouse 568 (1:500 in IF, Invitrogen, Cat. A-11004), goat anti-rabbit 488 (1:500 in IF, Invitrogen, Cat. A-11008), and goat anti-rat 568 (1:500 in IF, Invitrogen, Cat. A-11077). Nuclei were counterstained with DAPI (1:5000, Sigma, Cat. D9542). Stereology of aNSC samples and brain sections was performed as described previously, with minor modifications[48,49]. For brain sections, 1 out of every 5 sections was picked for immunostaining and 8–10 sections of each brain were adopted for stereology assay. All immunostainings were repeated at least three times, and representative images were displayed.

**Electroporation and luciferase assays.** Cultured WT and $Tet2^{-/-}$ aNSCs were used for electroporation assay with an electroporator (Amaxa Lonza). In brief, cell pellets were resuspended with 100 μl nucleofection solution. After electroporation, cells were plated into poly-ornithine/Laminin-coated 24-well plates with fresh proliferation medium. The second day, the proliferation medium was replaced with differentiation medium, and cells were continuously cultured for another 36–40 h. The luciferase assay was performed with a luminometer according to the manufacturer's protocol (Promega). A measure of 0.1 μg Renilla-luc plasmids and 2 μg NeuroD1-/GFAP-luc plasmids were used, respectively. For re-expression of Tet2, shRNA against Tet1 and Tet2 were obtained from Qiagen. For the acute

knockdown electroporation assay, the final concentration of siRNA was 100 nM (siGenome SMART Pool, M-058965-01, Dharmacon).

**Genomic DNA isolation and dot blot.** Cell pellets or brain tissues were homogenized in 300–600 μl DNA lysis buffer (100 mM Tris-HCL, pH = 8.0, 5 mM EDTA, 0.2% SDS and 200 mM NaCl) with Proteinase K freshly added, and incubated at 55 °C overnight. An equal volume of phenol:chloroform:isoamyl alcohol [25:24:1, saturated with 10 mM Tris (pH 8.0) and 1 mM EDTA; P-3803, Sigma] was added on the second day, mixed completely, and centrifuged at top speed on desktop centrifugation for 10 min. Supernatant was then mixed with an equal volume of isopropanol to precipitate DNA. DNA pellets were washed with 70% ethanol and eluted with 10 mM Tris–HCl (pH 8.0).

5-hmC-specific dot blots were performed using a Bio-Dot Apparatus (Bio-Rad) as described previously[18]. Briefly, DNA was treated with extensive RNase to ensure the complete removal of RNA contamination, and then spotted on an Amersham Hybond-N + membrane (GE Healthcare). DNA was fixed to the membrane by Stratagene UV Stratalinker 2400 (auto-crosslink). The membrane was blocked with 5% BSA and incubated with 5hmC antibody (Active Motif, Cat. No. 39791, 1:2000) overnight at 4 °C. Horseradish peroxidase–conjugated antibody to rabbit (1:5000, #A-0545, Sigma) was used as a secondary antibody, and incubated for 45 min at 20–25 °C. The signal was visualized by enhanced chemiluminescence. The density of each dot signal was quantified by ImageJ software. Unpaired t-test was used and error bars represented mean ± s.e.m.

**Quantitative RT–PCR.** Five hundred nanograms of total RNA isolated by TRIZOL was reverse transcribed using random hexamers to generate first-strand cDNA with SuperScript III according to the manufacturer's instructions (ThermoFisher). A volume of 1 μl of cDNA was used directly in 20 μl SYBR Green real-time PCR reactions that consisted of 1 × Power SYBR Green Master Mix, 0.5 μM forward and reverse primers, and nuclease-free water. 18S rRNA was used as an endogenous control for all samples, with 1 μl of cDNA diluted 1:10 in the nuclease-free water used. Reactions were run on an Applied Biosystems SDS 7500 Fast Instrument using the Standard 7500 default cycling protocol. Primers were designed using Primer3 online tools (http://biotools.umassmed.edu/bioapps/primer3_www.cgi).

**Biochemical immunoprecipitation.** For co-IP assays, WT aNSCs or 293FT cells transfected with the expression plasmids of FLAG-Foxo3a (Plasmid #8360, Addgene) and c-Myc-Tet2-FL (FL, full length). The cell pellets were lysed for 30 min on ice in RIPA Lysis and Extraction Buffer (Thermo) with protease inhibitor cocktail (Roche), sonicated briefly, and then centrifuged at 4 °C at maximal speed for 10 min. The supernatants were then incubated with Foxo3a antibody (NSCs) and ANTI-FLAG M2 Affinity Gel (Sigma-Aldrich, Cat. A4596) or EZview Red Anti-c-Myc Affinity Gel (Sigma-Aldrich, Cat. E6654) for 293 cells overnight. Mouse IgG-Agarose (Sigma-Aldrich) was used as control IgG. For DNase and RNase treatments, the supernatants of cell lysates were either treated with 30 units/ml of RQ1 RNase-Free DNase (Promega) or 25 μg ml$^{-1}$ RNase A for 20 min at 37 °C before IP. The beads were washed extensively with RIPA Lysis and Extraction buffer 3 times. The immunoprecipitates were eluted with 2 × Laemmli Sample Buffer (Bio-Rad), and the following immunoblotting assays were carried out to detect the target proteins as indicated. Anti-FLAG (1:1000 for western; Sigma-Aldrich, Cat. F1804), anti-c-Myc (1:1,000 for western, ThermoFisher, Cat. R950-25), anti-Foxo3a (1:1000 for western, Bethyl, Cat. A300-453A), and anti-Tet2 (1:500 for western, Diagenode, Cat. C15200179) antibodies were used. Images have been cropped for presentation. Full size images are presented in Supplementary Fig. 15.

**Chromatin immunoprecipitation.** Five to 10 million aNSCs were treated with 1% formaldehyde for 10 min at room temperature with gentle shaking. Fixation was terminated by adding 2 M glycine to reach a 0.125 M final concentration, and then shaking continued for an additional 5 min. Cells were collected by cell scrapers and spun down at 1,500 r.p.m. for 10 min. The cell pellet was resuspended in 1 ml Nuclei Swelling Buffer (10 mM HEPES/pH 7.9, 0.5% NP-40, 1.5 mM MgCl$_2$, 10 mM KCl, 0.5 mM DTT, and protease inhibitor cocktail), incubated on ice for 10 min, and centrifuged at 5,000 r.p.m. for 5 min. Nuclear pellets were further lysed in 1 ml SDS lysis buffer (20 mM HEPES/pH 7.9, 25% glycerol, 0.5% NP-40, 0.5% Triton X-100, 0.42 M NaCl, 1.5 mM MgCl$_2$, 0.2 mM EDTA, and protease inhibitor cocktail). Cell lysate was sonicated five times with 6–9 W output to obtain DNA fragments between 200 and 500 bp. After sonication, nuclear lysate was cleared by centrifugation at 13,000 r.p.m. for 10 min to keep supernatant. The nuclear lysate was diluted with 3–4 volumes of dilution buffer (0.01% SDS, 1% Triton X-100, 1.2 mM EDTA, 167 mM NaCl, 16.7 mM Tris-HCl/pH 8.0, and protease inhibitor cocktail). Immunoprecipitation was performed with antibodies specific to Foxo3a (1:1000, Bethyl) rotating overnight at 4 °C. 20 μl Protein-G Dynabeads (Life Technologies) were added for an additional 2 h. TSE I (0.1% SDS, 1% Triton X-100, 2 mM EDTA, 150 mM NaCl, 20 mM Tris-HCl/pH 8.1), TSE II (0.1% SDS, 1% Triton X-100, 2 mM EDTA, 500 mM NaCl, 20 mM Tris-HCl/pH 8.1), TSE III (0.25 M LiCl, 1% NP-40, 1% deoxycholate, 1 mM EDTA, 10 mM Tris-HCl/pH 8.1) were then stepwised applied to wash the beads to remove non-specific protein and

RNA binding. Beads were then washed twice with TE buffer before being eluted with 1% SDS and 0.1 M NaHCO$_3$. The elution was incubated at 65 °C for at least 6 h to reverse the formaldehyde cross-linking, and then DNA fragments were purified using a PCR Purification Kit (Qiagen). ChIPed DNA was subjected to library preparation.

**Genome-wide 5hmC profiling (hME-Seal).** hME-Seal was described previously[18]. Briefly, genomic DNA was first sonicated to 100–500 bp and mixed with 100 ml solution containing 50 mM HEPES buffer (pH 7.9), 25 mM MgCl$_2$, 250 mM UDP-6-N3-Glu, and 2.25 mM wild-type β-glucosyltransferase. Reactions were incubated for 1 h at 37 °C. DNA substrates were purified via Qiagen DNA purification kit. 150 mM dibenzocyclooctyne modified biotin was then added to the purified DNA, and the labeling reaction was performed for 2 h at 37 °C. The biotin-labelled DNA was enriched by Streptavidin-coupled Dynabeads (Dynabeads MyOne Streptavidin T1, Life Technologies) and purified by Qiagen DNA purification kit for library preparation.

**Foxo3a DNA probe binding assays.** dsDNA probes with either unmodified cytosine or 5hmC were independently mobilized on Dynabeads Streptavidin MyOne C1 (Life Technologies) with 1 × W/B buffer containing 25 mM Tris-Cl (pH 7.5), 1 mM EDTA, and 1M NaCl at room temperature for 30 min with gentle rotation. The beads were then immobilized with probes and blocked with Blocker BSA (10 ×), Thermo Fisher Scientific (Catalog number: 37520) in TBS for 30 min at room temperature. Beads were washed twice with 1x lysis buffer. The FOXO3A recombinant protein (Novus Biologicals, Cat. H00002309-P01) was added to blocked beads and incubated for 1 h at room temperature with gentle rotation. The beads were then washed extensively with 1x lysis buffer 5 times. FOXO3A was eluted in 2 × Laemmli Sample Buffer (Bio-Rad) at 100 °C for 10 min, and loaded onto an SDS-polyacrylamide gel for western blot to detect binding efficiency.

**Library preparation and high-throughput sequencing.** Enriched DNA from ChIP and hME-seal were subjected to library construction using the NEBNext ChIP-Seq Library Prep Reagent Set for Illumina according to the manufacturer's protocol. In brief, 25 ng of input genomic DNA or experimental enriched DNA were used for each library construction. 150–300-bp DNA fragments were selected by AMPure XP Beads (Beckman Coulter) after the adapter ligation. An Agilent 2100 BioAnalyzer was used to quantify the amplified DNA, and qPCR was applied to accurately quantify the library concentration. 20 pM diluted libraries were used for sequencing. 50-cycle single-end sequencings were performed using Illumina HISeq 2000. Image processing and sequence extraction were done using the standard Illumina Pipeline.

RNA-seq libraries were generated from duplicated samples per condition using the Illumina TruSeq RNA Sample Preparation Kit v2 following the manufacturer's protocol. The RNA-seq libraries were sequenced as 50-cycle paired-end runs using Illumina HiSeq 2000. A sequencing quality report was provided as Supplementary data 5.

**Bioinformatics analyses.** Bioinformatics analysis for ChIP-seq and 5hmC-seq were described previously[17,50]. Briefly, FASTQ sequence files were aligned to mm9 reference genome using Bowtie[51]. Peaks were identified by Model-based Analysis of ChIP-Seq (MACS) software[52]. Unique ChIP-seq and 5hmC-seq mapped reads were plotted to various genomic regions by ngsplot[53] or R (http://www.r-project.org/). Annotation analyses were performed by HOMER[54]. Gene ontology analysis was performed using Gene Ontology Consortium[55] or DAVID (the database for annotation, visualization and integrated discovery) bioinformatics resources[56]. RNA-seq reads were aligned using TopHat v2.0.13 (ref. 57), and differential RPKM expression values were extracted using cuffdiff v2.2.1 (ref. 57).

**Sample size and statistical analyses.** Investigators were always blinded to groups/genotypes for all experiments. All data are expressed as means ± s.e.m. GraphPad Prism (GraphPad Software Inc., La Jolla, CA, USA) was used for statistical analysis. Student's $t$-test was used to determine the differences between 2 groups with at least 3 replicates; 2-way analysis of variance followed by Bonferroni multiple-comparison test was used to compare replicate means if there was a significant difference between groups. $P < 0.05$ was considered statistically significant. For all microscopy analysis, the investigator was blinded to the experimental conditions. Binomal tests were performed in R.

**Data availability.** Data that support the findings of this study have been deposited in Gene Expression Omnibus (GEO) with the accession number GSE65994.

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

## Acknowledgements

We thank C. Strauss for critical reading of the manuscript and the members of the Jin lab for their assistance. X.L. was supported by the National Natural Science Foundation of China (31371309) and National Key Basic Research Program of China (No. 2014CB943001). P.J. was supported in part by NIH grants (NS051630, NS097206, NS079625, HG008935 and MH102690) and the Simons Foundation Autism Research Initiative (239320).

## Authors contributions

X.L., B.Y. and P.J. conceived and designed the project. X.L., B.Y., Y.K., Y.L., Y.C., L.L., Z.W., and M.W. performed the experiments. B.Y. performed the bioinformatics analyses. L.C. assisted with the computational coding. F.P., Q.D., Q.S., C.H., M.X. contributed the reagents. H.W., and Z.Q. supervised the statistical analyses. B.Y. and P.J. wrote the manuscript. All authors commented on the manuscript.

## Additional information

**Competing interests:** The authors declare no competing financial interests.

