## [Peer Review File · Nature Communications]

Reviewers' comments:

Reviewer #1 (Remarks to the Author):

In this manuscript, the authors demonstrated that Tet2 regulates the proliferation and differentiation of adult neural stem cells (aNSC) through interaction with Foxo3a, and that the Tet2-Foxo3a axis is critical for controlling genes related with adult neurogenesis through epigenetic modification. To make their conclusions more convincing, there are a few points that need to be addressed.

Major comments

1. The authors showed in Fig. 1e that when WT aNSCs were induced into differentiation, the expression of Tet1 was reduced whereas the expression of Tet2 was up-regulated. Consistent with the opposite trend of gene expression of Tet1 and Tet2 during the switch from aNSC proliferation to differentiation, knockout of Tet2 led to increased proliferation and decreased differentiation of aNSCs, in a manner opposite to knockout of Tet1. It will be very interesting to know what will happen to aNSC proliferation and differentiation under Tet1 and Tet2 double knockout condition.

2. In Fig. 2, the evidence showing that Tet2 regulates adult neurogenesis need to be strengthened.

a. In Fig. 2a-d, Nestin/BrdU and/or Sox2/BrdU double staining would be a better index of NSC proliferation than BrdU staining alone.

b. In Fig 2q to Fig 2t, representative staining images from both WT and Tet2^{-/-} mice should be shown, in addition to showing the quantification data. In Supplementary Fig. 4, BrdU staining in the DG of both WT and Tet2^{-/-} mice should be shown, instead of just BrdU staining in the WT DG.

c. For Fig. 2q, Ki67 staining should be shown to strengthen the conclusion of Tet2 in regulating the proliferation of aNSC, in addition to BrdU staining. Moreover, Nestin/BrdU and/or Sox2/BrdU double staining instead of BrdU single staining should be shown.

d. The authors showed in vivo neurogenesis in SGZ region. How about SVZ region?

3. Because the main conclusion of the manuscript is that Tet2 interacts with Foxo3a to regulate adult neurogenesis, it would be nice if the authors provide evidence showing that Foxo3a works with Tet2 to regulate proliferation and differentiation of aNSC in vitro and in vivo (ideally). At least, proliferation and differentiation of aNSC in Tet2^{-/-} aNSC with Foxo3a knockdown should be analyzed.

4. Several major questions need to be addressed regarding the model (Fig 7).

a. In the model (Fig 7), under Tet2 deletion condition, how Foxo3a will be increasingly recruited to the promoter region of the genes commonly regulated by Foxo3a and Tet2 is not clear. Does the interaction of Tet2 and Foxo3a play a role here? And how?

b. What's the cellular function of interaction between Tet2 and Foxo3a? Is the interaction important for controlling the transcription of regulated genes? And how?

c. If interaction between Tet2 and Foxo3a is important for their function in controlling gene expression, in Fig 5a, showing co-localization of Foxo3a binding sites and Tet2 binding sites would be more convincing than showing overlapping of Foxo3a peaks and 5hmC peaks.

Minor questions:

1. Fig. 1h & 1i can be put into supplementary figure.

2. What is the difference between Fig. 1g and Supplementary Fig 2a?

3. The description in text did not match the data showing in the figure. "The aNSCs from the Tet2^{-/-} mice retained the ability to differentiate into neuronal and glial cells (Supplementary Fig.2c-f)." Supplementary Fig. 2c-f only showed Tet2^{-/-} NSCs express Nestin and Sox2, did not show the ability of these cells to differentiate.

4. For Supplementary Fig.3 c,d,e,f, the author showed that the expression of Tuj1, NeuroD1, GFAP s100 β in Tet2^{-/-} aNSCs is decreased, compared to WT aNSCs. The decreased expression of these genes is due to decreased percentage of cells expressing indicated markers or is due to decreased

expression of indicated markers in individual cells?

5. In Fig. 4b, the label in the figure is wrong. "5hmC distribution on 979 significantly upregulated genes" should be "5hmC distribution on 979 significantly downregulated genes".

6. In Fig. 5b, The Tet2 blot is not clear. It is not obvious there is a band corresponding to Tet2 in the Foxo3a IP lane, or just smear.

Reviewer #2 (Remarks to the Author):

The study from Li and colleagues examines the novel role of Ten-eleven translocation 2 (Tet2) in adult neurogenesis. Depletion of Tet2 leads to increased adult neural stem cell (aNSC) proliferation and reduced differentiation. The associated transcriptome and 5hmC alterations are herein profiled and demonstrate concomitant alterations. It also appears that a transcription factor Foxo3a interacts with Tet2 during this process to coordinate gene expression and 5hmC changes. This interesting study utilizes several different methodologies in order to identify a regulatory role for Tet2, together with Foxo3a, in adult neurogenesis. However, despite the wealth of data, the manuscript needs to be strengthened in order to be able to make an explicit and conclusive case. Below are my specific comments.

Major points:

1. The study mostly relies on a Tet2 knockout mouse line to probe the function of Tet2 in adult neurogenesis. Although a small change in Tet2 expression is reported from the RNAseq experiment, it is important to demonstrate the Tet2 knockout at the protein level. Also, can the authors replicate their findings (morphology, mRNA, 5hmC) via the Tet2 siRNA construct that they already have (e.g. in suppl. Fig. 3)? A more definitive conclusion can be drawn if the group can perform Tet2 overexpression to rescue the deficits reported in the Tet2 knockout.

2. I found the approach to the Tet2-Foxo3a interaction somewhat confusing. As Tet2 converts methylated cytosine into 5hmC, a Tet2 knockout/decrease should lower the 5hmC level, as shown in this study (Fig. 1f). Consistently, there are more down regulated transcripts (979) than upregulated ones (979 genes vs. 703 genes). However, the authors only focus their analyses on the upregulated genes and the 5hmC increase. The rationale behind this approach is their likelihood to reflect Tet2's direct targets (Pg. 12). But this is very subjective. Without more experimental evidence to support this assumption, the authors should include the 979 downregulated genes in their analysis. One benefit of this additional approach is that it could serve as a control to see if the findings from the upregulated genes are specific. Moreover, it should be explained clearly how "intragenic region", "promoter", and "gene body" are defined in this study. More specifically, if the "intragenic region" includes gene "promoter" in the analyses? Though there is an expected global 5hmC "decrease" after Tet2 knockout, the authors focus on the "intragenic" 5hmC "increase". Not much evidence is provided here to support an effect of Tet2 knockdown on intragenic 5hmC increase (ref 48). Since many of the 5hmC increases in the absence of Tet2 are localized to promoter regions (Fig. 4a,b), it would be of some benefit to discuss if this has previously been reported and what mechanisms could possibly be involved. It also appears that 5hmC is upregulated in promoters of both up and downregulated genes (Figs. 4a, 4b), whereas the gene body 5hmC has opposite direction of changes in the two groups. Are these changes illustrated in Figs. 4a,4b significant? To connect the promoter enrichment of Foxo3a with intragenic 5hmC alterations in the current form of analysis is difficult to interpret. In order to clarify this, the authors should redo their Foxo3a-5hmC analyses on promoter and non-promoter regions separately. They should also analyze the downregulated genes which have a promoter 5hmC increase and gene-body decrease to further strengthen the findings.

3. The authors performed various sequencing experiments (ChIP-seq, RNA-seq, 5hmC-seq), however they did not validate these hits despite an mRNA q-PCR on three candidate genes (Fig. 6 c-e). This is

a necessary control experiment commonly used to confirm the results of genome-wide screens. Likewise, it is also necessary to show the specificity of the antibodies used in the ChIP assay.

Additional points:

1. It would be useful if the gender information is stated for both in vivo and in vitro studies.
2. What are the relative expression levels of Tet1, 2, and 3 during adult neurogenesis? And does Tet1 or Tet3 change when Tet2 is manipulated? This is important for understanding the role of Tet2.
3. There are two aNSC culture conditions referred to in the manuscript - one is proliferating and the other is differentiated aNSC. These two states have different epigenetic status, including 5hmC levels. Though some assays indicate their explicit culture conditions, it is important to define if the aNSCs are differentiated or proliferating in each aNSC culture experiment throughout the manuscript.
4. I will recheck that this was not mentioned and I missed it, but a sequencing quality report should be included on all ChIP-seq, RNA-seq, and 5hmC-seq experiments.
5. The effectiveness of shRNAs at knocking down the target gene (e.g. Foxo3a, Tet2) should be shown.
6. Figure 1d is said to show localization of 5hmC in the nuclei of nestin positive aNSCs. But I don't believe the figure is sufficiently clear.
7. Figure 4b, Please change "upregulated" to "downregulated" on x-axis.
8. Figure 5d: what does the pyramid shape stand for?

Reviewer #3 (Remarks to the Author):

In this manuscript, "Ten-eleven translocation 2 interacts with Forkhead box O3 and regulates adult neurogenesis" by Li et al., the authors showed that depletion of Tet2 in aNSCs impairs and maintains the differentiation and proliferation statuses of aNSCs, respectively. They further suggested that a subset of gene expressions are regulated by the inter play between Tet2 and Foxo3. This manuscript is potentially interesting but its conceptual advance in this field seems to fall short of what is typically required to warrant publication in this journal.

Specific comments

1. Regarding, Sup Fig. 1, indicate % of Nestin+/Sox2+ cells among total cells (proliferating condition), and % of Nestin+ or Sox2+ cell in addition to Tuj1+ and GFAP+ cells in the differentiation condition. Is there any GFAP+/Nestin+ (dually positive) cells in the differentiation condition?
2. Page 7, line 9-11, they described "The aNSCs from the Tet2^{-/-} mice retained the ability to differentiate into neuronal and glial cells (Supplementary Fig. 2c-f)". However, they actually showed staining of only undifferentiated markers (Nestin and Sox2) in this figure. This is not appropriate.
3. In terms of Fig. 2, I think it would be better if the authors would provide the staining of Nestin or Sox2 and the quantification of the marker positive cells.
4. In Sup Fig. 3g and h, they showed both neuronal and glial gene promoters are suppressed by Tet2-KD. These promoters are from genes in opposite lineages. Explain how this could be possible simultaneously.
5. It is not clearly described how the authors come up with the idea of interplay between Tet2 and Foxo3, although both of them have individually known to be important in a variety of biological events.
6. As for Fig. 5, they need to show the data for the immunoprecipitation using the control IgG in addition to the data using anti-Foxo3a antibody.
7. Although the authors showed that Tet2 and Foxo3 associate each other, they also indicated that Foxo3a binding to the target genes were increased in Tet2^{-/-} aNSCs compared to WT cells. Discuss the mechanism.

Minor comments

1. Regarding Fig. 4b, "downregulated" (in the text) and "upregulated" (in the figure) are inconsistent. Check which one is correct.

Responses to Reviewers' Comments:

We would like to thank the reviewers for their thoughtful and constructive comments. Following their suggestions, we have carried out substantial amount of new experiments to address the concerns they raised. These additional experiments confirmed our original conclusions and significantly strengthened this manuscript. Below are the point-by-point responses to each reviewer.

Reviewer #1

We would like to thank this reviewer for the positive comments. We have performed additional experiments and provided discussion to address the reviewer's concerns.

1. *“The authors showed in Fig. 1e that when WT aNSCs were induced into differentiation, the expression of Tet1 was reduced whereas the expression of Tet2 was up-regulated. Consistent with the opposite trend of gene expression of Tet1 and Tet2 during the switch from aNSC proliferation to differentiation, knockout of Tet2 led to increased proliferation and decreased differentiation of aNSCs, in a manner opposite to knockout of Tet1. It will be very interesting to know what will happen to aNSC proliferation and differentiation under Tet1 and Tet2 double knockout condition.”*

This is an excellent question. A previous study (Zhang et al. 2013, Cell Stem Cell) demonstrated that loss of Tet1 diminishes the adult SGZ neural progenitor pool, which is, as described by the reviewer, opposite to the function of Tet2 in our current study. To address the reviewer's question, we have successfully established the Tet1 and Tet2 double knockout (dKO) mice. However, consistent with the observation by Jaenisch group (Dawlaty et al, 2013, Dev. Cell), the Tet1 and Tet2 double knockout results in substantial embryonic lethality, with most of the escapers die within two days after birth. Thus, it would be difficult to directly examine adult neurogenesis in Tet1/Tet2 double KO mice. As an alternative, we have performed Tet1 and Tet2 double knockdown (dKD) by shRNA in wildtype aNSCs and investigated their proliferation and differentiation states. Neuronal differentiation was assessed by the expression of NeuroD1 and GFAP using luciferase reporter assays as described in Supplementary Fig. 4a. We found that simultaneous knockdown of both Tet1 and Tet2 in wildtype aNSCs resulted in a significant decrease of both NeuroD1 and GFAP reporter expression, indicating the double-knockdown impaired aNSC differentiation and recapitulating the effects of *Tet2*^{-/-} and Tet2 single knockdown aNSCs (Supplementary Fig. 5c,d). On the other hand, co-depletion of both Tet1 and Tet2 significantly enhanced the aNSC proliferation rate, evidenced by significant increase of BrdU positive cells (Supplementary Fig. 5e,f). These data together suggest that the Tet1 and Tet2 double knockdown in adult NSCs can increase the proliferation and decrease the differentiation in WT NSCs, a similar effect we observed in *Tet2*^{-/-} aNSCs and Tet2 transient knockdown by shRNA (Supplementary Fig. 5a,b). To strengthen the direct roles of Tet2 in modulating adult neurogenesis, we performed electroporation to re-express Tet2 in the *Tet2*^{-/-} aNSCs. The re-expression of Tet2 significantly restored the NeuroD1 and GFAP reporter expression comparing to *Tet2*^{-/-} aNSCs, supporting the key role of Tet2 in adult neurogenesis (Supplementary Fig. 4i,j). Thus, these new data together strongly support the notion that Tet2 plays predominant epigenetic role in regulating adult neurogenesis.

2. *“In Fig. 2, the evidence showing that Tet2 regulates adult neurogenesis need to be strengthened.”*
a. *“In Fig. 2a-d, Nestin/BrdU and/or Sox2/BrdU double staining would be a better index of NSC proliferation than BrdU staining alone.”*

This is a great suggestion. We have performed the Nestin/BrdU and Sox2/BrdU co-staining in both WT and *Tet2*^{-/-} aNSCs (Supplementary Fig. 3a,b). Our new data suggested that all BrdU⁺ cells were co-stained with Nestin and Sox2. These results confirmed the BrdU staining faithfully indicated NSC proliferation state in our study.

b. “In Fig 2q to Fig 2t, representative staining images from both WT and *Tet2*^{-/-} mice should be shown, in addition to showing the quantification data. In Supplementary Fig. 4, BrdU staining in the DG of both WT and *Tet2*^{-/-} mice should be shown, instead of just BrdU staining in the WT DG.”

Based on the excellent suggestion from this reviewer, we have included the BrdU and Ki67 co-staining in the SGZ from one-week old mice (Supplementary Fig. 6a). There were significant higher BrdU⁺ cells and Ki67⁺ cells in *Tet2*^{-/-} mice. This observation was consistent with the notion that the SGZ is one of the main neurogenic regions *in vivo*, and confirmed *Tet2* depletion resulted in enhanced aNSC proliferation *in vivo*. At the 4-week time point, some newborn neurons migrated to the granular zone and became mature neurons (Supplementary Fig. 6b, NeuN staining), and total BrdU⁺ cells in *Tet2*^{-/-} mice were still significantly higher than in WT mice (Fig. 2r and Supplementary Fig. 6b, $p < 0.05$, unpaired *t*-test). However, there were significantly fewer (43% fewer) newborn neurons (BrdU⁺/NeuN⁺) in *Tet2*^{-/-} mice than in WT mice (Fig. 2s and Supplementary Fig. 6b, $p < 0.05$, unpaired *t*-test).

c. “For Fig. 2q, Ki67 staining should be shown to strengthen the conclusion of *Tet2* in regulating the proliferation of aNSC, in addition to BrdU staining. Moreover, Nestin/BrdU and/or Sox2/BrdU double staining instead of BrdU single staining should be shown.”

This is a good suggestion. As indicated above, we have performed BrdU-Ki67 co-staining in WT and *Tet2*^{-/-} mice SGZ *in vivo*, and both proliferation markers showed increased staining in *Tet2*^{-/-} mice (Supplementary Fig. 6a). We have attempted multiple times to perform the Nestin/BrdU and Sox2/BrdU co-staining *in vivo*, but failed to obtain good quality images. Nonetheless, the Nestin/BrdU and Sox2/BrdU co-staining in both WT and *Tet2*^{-/-} aNSCs (Supplementary Fig. 3a,b) confirmed BrdU staining faithfully indicated NSC proliferation state in our study.

d. “The authors showed *in vivo* neurogenesis in SGZ region. How about SVZ region?”

To address this question, we investigated the SVZ neurogenesis by BrdU injection as described in the methods, followed by BrdU-Ki67 co-staining in both WT and *Tet2*^{-/-} mice SVZ. A significant increase of BrdU positive cells were found in the SVZ of *Tet2*^{-/-} mice than WT, indicating the depletion of *Tet2* can also promote SVZ neurogenesis (Supplementary Fig. 7a, b, $p < 0.001$, unpaired *t*-test).

3. “Because the main conclusion of the manuscript is that *Tet2* interacts with *Foxo3a* to regulate adult neurogenesis, it would be nice if the authors provide evidence showing that *Foxo3a* works with *Tet2* to regulate proliferation and differentiation of aNSC *in vitro* and *in vivo* (ideally). At least, proliferation and differentiation of aNSC in *Tet2*^{-/-} aNSC with *Foxo3a* knockdown should be analyzed.”

We thank the reviewer for this excellent suggestion. To test the direct role of *Foxo3a* in gene regulation related to aNSC proliferation, we first investigated several upregulated genes from Fig. 6a and 6b. *Tet2* depletion significantly elevated the transcription of these genes, consistent with RNA-seq results. Importantly, shRNA knockdown of *Foxo3a* in the *Tet2*^{-/-} aNSCs reduced the elevated transcription of these genes approximately to the level of WT aNSCs, confirming that *Foxo3a* directly modulate these upregulated genes related to aNSC proliferation caused by the loss of *Tet2* (Fig. 6c-e, unpaired *t*-test). Since *Foxo3a* directly regulate genes involved in proliferation, we then investigated the cell proliferation states in control, *Tet2*^{-/-} and *Foxo3a* knockdown in the *Tet2*^{-/-} aNSCs by BrdU staining. Consistently, the *Foxo3a* knockdown in the *Tet2*^{-/-} aNSCs significantly reduced the proliferation, supporting a direct role of *Foxo3a* in regulating genes involved in aNSC proliferation (Fig. 6f).

4. “Several major questions need to be addressed regarding the model (Fig 7).”

a. “In the model (Fig 7), under *Tet2* deletion condition, how *Foxo3a* will be increasingly recruited to

the promoter region of the genes commonly regulated by Foxo3a and Tet2 is not clear. Does the interaction of Tet2 and Foxo3a play a role here? And how?"

In the present study, we found that the loss of Tet2 in aNSCs disrupted their proliferation and differentiation, and caused impaired neurogenesis *in vivo* and *in vitro*. At molecular level, differential expressed (DE) genes in response to Tet2 depletion correlated with our phenotypical observations. These data suggested maintaining proper expression levels of these genes by Tet2 are critical for the aNSC identify and functions. As a transcription activator, we show that Foxo3a physically interacts with Tet2 and maintains the expression homeostasis of these DE genes. Interestingly, the panel of upregulated genes in *Tet2*^{-/-} aNSCs appeared to be explicitly associated with proliferating aNSCs, as these genes did not show obvious differences in the differentiated WT or *Tet2*^{-/-} aNSCs (Fig. 3c, Proliferating versus Differentiation). We found a substantial and significant increase of Foxo3a occupancy on the promoters of these upregulated genes when Tet2 was removed (Supplementary Fig. 12), indicating its direct role in driving the overexpression of these genes. Consistently, we also found the knockdown of Foxo3a in the *Tet2*^{-/-} background could rescue the overexpression of these genes due to the absence of Tet2 (Fig. 6c-e).

The reviewer raised an excellent question that how Foxo3a will be increasingly recruited to the promoter regions. In fact, there is a positive correlation between increased Foxo3a occupancy and elevated 5hmC in promoters of these upregulated genes in the absence of Tet2 (Supplementary Fig. 12). We speculated this in the discussion of our initial submission that, the presence of Tet2 on these genes could maintain the precise 5hmC homeostasis, which could determine the Foxo3a binding dynamics to keep proper amount of Foxo3a occupancy to control precise gene expression during neurogenesis. Thus, loss-of-Tet2 results in aberrant accumulation of 5hmC, possibly due to the impaired 5hmC oxidation process (5hmC to 5fC/5caC) and recruits additional Foxo3a to ectopically increase the expression of these genes.

To draw the link between 5hmC accumulation and increased Foxo3a occupancy, we first examined the Foxo3a and 5hmC dynamic changes on the promoters of both significantly upregulated and downregulated genes in the absence of Tet2 (Supplementary Fig. 13a). Importantly, both Foxo3a and 5hmC showed concomitantly stronger enrichment on the promoters of upregulated genes than downregulated genes when Tet2 is depleted (Supplementary Fig. 13a). These observations support the notion that the aberrant accumulation of 5hmC upon Tet2 depletion could further recruit substantial more Foxo3a to the promoters of upregulated genes to stimulate their expression. To test the direct interaction between 5hmC and Foxo3a, we synthesized two sets of biotin-labelled DNA oligos with identical sequences based on Foxo3a ChIP-seq motifs. The cytosines in the consensus sequence were either hydroxymethylated (5hmC probe) or unmodified (control probe), with the latter used as a negative control. Control and 5hmC-containing probes were used for *in vitro* binding by adding with equal amount of Foxo3a recombinant protein. We found that Foxo3a showed significant stronger binding to the 5hmC-modified probes over control *in vitro* (Supplementary Fig. 14a,b). These data, together with Foxo3a/5hmC correlation *in vivo*, revealed the molecular mechanism of upregulated genes that 5hmC accumulation upon Tet2 deletion served as a binding platform to further recruit Foxo3a and stimulate gene expression.

In summary, here we uncovered a positive correlation between significantly increased 5hmC and Foxo3a occupancy specifically on the promoters of these upregulated genes, which explained the mechanism of the upregulation of these genes at the molecular level. These data presented a potentially intriguing negative feedback loop of Tet2-Foxo3a axis. During normal aNSC state, Tet2 interacted with Foxo3a to modulate proper expression level of these proliferation-related genes. While Foxo3a is the direct transcription modulator, Tet2 could actively promote the conversion of promoter 5hmC to its

downstream derivatives, which ensure the ideal epigenetic environment to land proper amount of Foxo3a. The absence of Tet2 resulted in the accumulation of 5hmC on the promoters, which could ectopically recruit excessive Foxo3a that led to uncontrolled overexpression of these proliferation related genes (Fig. 7).

b. *“What's the cellular function of interaction between Tet2 and Foxo3a? Is the interaction important for controlling the transcription of regulated genes? And how?”*

This is another excellent question raised by this reviewer. As explained above, there could be a potentially intriguing negative feedback loop of Tet2-Foxo3a axis. During normal aNSC state, Tet2 interacted with Foxo3a to modulate proper expression level of these proliferation-related genes. While Foxo3a is the direct transcription modulator, Tet2 could actively promote the conversion of promoter 5hmC to its downstream derivatives, which ensure the ideal epigenetic environment to land proper amount of Foxo3a. Thus, the Tet2-Foxo3a interaction will be critical to co-exist on these loci for gene regulation.

Furthermore, it is plausible that Foxo3a could contribute the Tet2 to target these genes involved in neurogenesis, especially given the fact that Tet2 is lacking DNA-binding domain *per se*. Moreover, a very recent study in mESC suggested that another transcription factor, SALL4A, could directly bind to 5hmC, and coordinate with Tet2 to stimulate further oxidation of 5hmC (from 5hmC to 5fC/5caC) (Xiong et al, 2016, Mol. Cell). SALL4A, in many ways, resembles the characteristics of Foxo3a, such as binding to 5hmC-modified DNA as well as coordinating with Tet2. Thus, it is possible that the Foxo3a possesses similar roles to function as Tet2 co-factor to modulate its enzymatic activity. It will be very interesting to pursue these possibilities in the future, but these are beyond the scope of the current study. We have included this point to our discussion.

c. *“If interaction between Tet2 and Foxo3a is important for their function in controlling gene expression, in Fig 5a, showing co-localization of Foxo3a binding sites and Tet2 binding sites would be more convincing than showing overlapping of Foxo3a peaks and 5hmC peaks.”*

We agree with this reviewer's comment, and have attempted numerous times to perform the Tet2 ChIP-seq in aNSCs with various Tet2 antibodies from different vendors. However, the lack of Tet2 ChIP grade antibody prevented us from obtaining such data. We have requested an in-house Tet2 antibody from a publication (Chen et al, 2013, Nature), but failed to reproduce their Tet2 ChIP-seq data.

5. *“Fig. 1h & 1i can be put into supplementary figure.”*

This data demonstrated a specific accumulation of 5hmC in aNSCs upon differentiation to neuronal and glial cells. In contrast, mouse embryonic stem cells showed significant reduction of 5hmC when differentiate to embryonic bodies. Thus, our present data is highly significant to identify the epigenetic mechanism related to adult neurogenesis.

6. *“What is the difference between Fig. 1g and Supplementary Fig 2a?”*

We apologize for the confusion. Fig.1g showed the 5hmC quantification in proliferating and differentiated WT and *Tet2*^{-/-} aNSCs from 5hmC dot blots. Supplementary Fig 2a showed the 5hmC quantification in WT and *Tet2*^{-/-} aNSCs by HPLC. We have described it more clearly in the revised manuscript.

7. *“The description in text did not match the data showing in the figure. “The aNSCs from the Tet2^{-/-} mice retained the ability to differentiate into neuronal and glial cells (Supplementary Fig.2c-f).”*

Supplementary Fig. 2c-f only showed Tet2^{-/-} NSCs express Nestin and Sox2, did not show the ability of these cells to differentiate.”

We apologize for the error. We intended to demonstrate the *Tet2^{-/-}* NSCs still possessed NSC identity with positive Nestin and Sox2 staining here. The impaired neurogenesis in *Tet2^{-/-}* aNSCs was demonstrated in main Fig. 2f-2m in the next section. We have revised the text to correct this.

8. *“For Supplementary Fig.3 c,d,e,f, the author showed that the expression of Tuj1, NeuroD1, GFAP s100 β in Tet2^{-/-} aNSCs is decreased, compared to WT aNSCs. The decreased expression of these genes is due to decreased percentage of cells expressing indicated markers or is due to decreased expression of indicated markers in individual cells?”*

This is a good question. Based on Fig. 2f-2m, there were fewer Tuj1 and GFAP positive cells in *Tet2^{-/-}* aNSCs, and the IF intensity was weaker. Thus, we conclude the decreased expression of these genes could be due to both decreased percentage of cells expressing indicated markers, as well as the decreased expression of indicated markers in their positive cells. This notion was further confirmed by NeuN staining *in vivo* (Supplementary Fig. 6b).

9. *“In Fig. 4b, the label in the figure is wrong. “5hmC distribution on 979 significantly upregulated genes” should be “5hmC distribution on 979 significantly downregulated genes”.”*

We apologize for the error and have corrected it in the revised manuscript.

10. *“In Fig. 5b, The Tet2 blot is not clear. It is not obvious there is a band corresponding to Tet2 in the Foxo3a IP lane, or just smear.”*

We have included a new IP-western blot along with IgG control to clearly demonstrate the association between Foxo3a and Tet2 endogenous proteins in the revised manuscript (Fig. 5b).

Reviewer #2

We would like to thank this reviewer for recognizing the importance of our study and the wealth of data that we have presented in this manuscript. We have carefully considered the comments from this reviewer and performed additional experiments to address these concerns.

1. *“The study mostly relies on a Tet2 knockout mouse line to probe the function of Tet2 in adult neurogenesis. Although a small change in Tet2 expression is reported from the RNAseq experiment, it is important to demonstrate the Tet2 knockout at the protein level. Also, can the authors replicate their findings (morphology, mRNA, 5hmC) via the Tet2 siRNA construct that they already have (e.g. in suppl. Fig. 3)? A more definitive conclusion can be drawn if the group can perform Tet2 overexpression to rescue the deficits reported in the Tet2 knockout.”*

This is an excellent question. As included in the initial submission, the *Tet2^{-/-}* mice were generated by replacing part of exon 3 sequences of the Tet2 gene with nlacZ/nGFP41 (Li et al, 2011, Blood). Thus, the RNA-seq would not be ideal to distinguish the *Tet2^{-/-}* with control. To fully address the reviewer’s comment, we have now provided the Western blot to demonstrate the loss of Tet2 protein in the *Tet2^{-/-}* mice (Supplementary Fig. 10c).

In order to validate the key roles of Tet2 in adult neurogenesis, we transiently knocked down Tet2 in wildtype aNSCs by shRNA electroporation (Supplementary Fig. 5a). The Tet2 knockdown led to the upregulation of Nestin and downregulation of NeuroD1 and GFAP, consistent with the enhanced proliferation and impaired differentiation in *Tet2^{-/-}* aNSCs (Supplementary Fig. 5b). These observations ruled out the indirect effects caused by Tet2 germline deletion and support the direct and genuine roles of Tet2 in adult neurogenesis. Importantly, simultaneous knockdown of both Tet1 and

Tet2 in wildtype aNSCs resulted in a significant decrease of both NeuroD1 and GFAP reporter expression, indicating the double-knockdown impaired aNSC differentiation and recapitulating the effects of *Tet2*^{-/-} and *Tet2* single knockdown aNSCs (Supplementary Fig. 5c,d). On the other hand, co-depletion of both *Tet1* and *Tet2* significantly enhanced the aNSC proliferation rate, evidenced by significant increase of BrdU positive cells (Supplementary Fig. 5e,f). These observations indicated the *Tet1* and *Tet2* double knockdown in adult NSCs fully resembled the characteristics comparing to *Tet2*^{-/-} aNSC. Collectively, our data support the predominant roles of *Tet2* in regulating adult neurogenesis, and the loss of *Tet2* shifted the balance of aNSC states by promoting proliferation and impairing their differentiation.

We do agree with the reviewer that “a more definitive conclusion can be drawn if we could perform *Tet2* overexpression to rescue the deficits reported in the *Tet2*^{-/-}.” To address this comment, we performed electroporation to re-express *Tet2* in the *Tet2*^{-/-} aNSCs. The re-expression of *Tet2* significantly restored the NeuroD1 and GFAP reporter expression comparing to *Tet2*^{-/-} aNSCs (Supplementary Fig. 4i,j). These data provide further support of a critical and direct role of *Tet2* in adult neurogenesis.

2. *“I found the approach to the Tet2-Foxo3a interaction somewhat confusing. As Tet2 converts methylated cytosine into 5hmC, a Tet2 knockout/decrease should lower the 5hmC level, as shown in this study (Fig. 1f). Consistently, there are more down regulated transcripts (979) than upregulated ones (979 genes vs. 703 genes). However, the authors only focus their analyses on the upregulated genes and the 5hmC increase. The rationale behind this approach is their likelihood to reflect Tet2's direct targets (Pg. 12). But this is very subjective. Without more experimental evidence to support this assumption, the authors should include the 979 downregulated genes in their analysis. One benefit of this additional approach is that it could serve as a control to see if the findings from the upregulated genes are specific. Moreover, it should be explained clearly how "intragenic region", "promoter", and "gene body" are defined in this study. More specifically, if the "intragenic region" includes gene "promoter" in the analyses? Though there is an expected global 5hmC "decrease" after Tet2 knockout, the authors focus on the "intragenic" 5hmC "increase". Not much evidence is provided here to support an effect of Tet2 knockdown on intragenic 5hmC increase (ref 48). Since many of the 5hmC increases in the absence of Tet2 are localized to promoter regions (Fig. 4a,b), it would be of some benefit to discuss if this has previously been reported and what mechanisms could possibly be involved. It also appears that 5hmC is upregulated in promoters of both up and downregulated genes (Figs. 4a, 4b), whereas the gene body 5hmC has opposite direction of changes in the two groups. Are these changes illustrated in Figs. 4a,4b significant? To connect the promoter enrichment of Foxo3a with intragenic 5hmC alterations in the current form of analysis is difficult to interpret. In order to clarify this, the authors should redo their Foxo3a-5hmC analyses on promoter and non-promoter regions separately. They should also analyze the downregulated genes which have a promoter 5hmC increase and gene-body decrease to further strengthen the findings.”*

We thank the reviewer to raise the issue that our terminology is not clearly defined. We defined 500 bp upstream and downstream of transcription start sites (TSS) as promoter regions, and we have re-defined the gene bodies by subtracting the 500bp downstream of TSS from full length transcripts, thus the gene bodies refer to the region between 500bp downstream of TSS and transcription end sites (TES). On the other hand, we refer the whole transcripts from TSS to TES as intragenic regions. We apologize for the confusion and have systematically corrected these terminologies in the text.

Per reviewer's request, we re-calculated the *Foxo3a* and 5hmC dynamic changes on the promoters and gene bodies of overall 310 upregulated genes bound by *Foxo3a* (Supplementary Fig. 12), and two subsets of genes enriched in cell cycle and nervous development pathways (Fig. 6a,b).

The new analyses in overall upregulated genes showed consistent results in our initial submission that, Foxo3a showed higher enrichment on promoters than gene bodies (Foxo3a promoters vs gene bodies, color intensity), and Tet2 depletion led to significantly increased Foxo3a binding to these promoters (violin plots, $p < 0.0001$, unpaired t-test). 5hmC levels also became significantly increased in these promoters in the absence of Tet2 (violin plots, $p < 0.0001$, unpaired t-test). Both Foxo3a and 5hmC on gene bodies showed a modestly significant increase in *Tet2*^{-/-} versus WT (violin plots, $p < 0.01$, unpaired t-test). Similarly, the analyses on the two sub-groups of genes displayed consistent results that Foxo3a showed the most significant elevation on promoters of genes in both groups (violin plots, $p < 0.01$, unpaired t-test), again correlated with promoter 5hmC dynamics (Fig. 6a,b). These new analyses confirmed our conclusion that, the functional interplay between Tet2 and Foxo3a regulates aNSC genes involved in proliferation and aNSC signature, and depletion of Tet2 resulted in a concurrent increase of Foxo3a and 5hmC on these promoters to ectopically elevate these genes. Consistently, we found knockdown of Foxo3a in the *Tet2*^{-/-} background rescued the overexpression of these genes due to the absence of Tet2 (Fig. 6c-e). Since Foxo3a directly regulate genes involved in proliferation, we then investigated the cell proliferation states in control, *Tet2*^{-/-} and Foxo3a knockdown in the *Tet2*^{-/-} aNSCs by BrdU staining. Consistently, the Foxo3a knockdown in the *Tet2*^{-/-} aNSCs significantly reduced the proliferation, providing solid evidence on the direct roles of Foxo3a in regulating genes involved in aNSC proliferation (Fig. 6f). We further tested the direct interaction between 5hmC and Foxo3a by *in vitro* Foxo3a-5hmC/control probe binding assays. We found that Foxo3a showed significant stronger binding to the 5hmC-modified probes over control *in vitro* (Supplementary Fig. 14a,b). These data together revealed the molecular mechanism of upregulated genes that 5hmC accumulation upon Tet2 deletion served as a bait to further recruit Foxo3a to stimulate these gene expression.

To validate the predominant roles of Foxo3a to regulate upregulated genes in *Tet2*^{-/-} aNSCs, we compared the Foxo3a and 5hmC dynamic changes on the promoters and gene bodies of significantly upregulated and downregulated genes (Supplementary Fig. 13a). Both Foxo3a and 5hmC showed substantially more and significant enrichment on the promoters of upregulated genes than downregulated genes in *Tet2*^{-/-} aNSCs comparing to WT (Supplementary Fig. 13a). These observations strongly support the predominant roles of Foxo3a on upregulated genes by binding to their promoters. Overall Foxo3a and 5hmC changes on the gene bodies positively correlated with their expression, but the Foxo3a changes on the gene bodies of downregulated genes were not statistically significant (Supplementary Fig. 13b, $p = 0.2$, unpaired t-test). These data suggested the positive correlation of Foxo3a and 5hmC primarily for promoter binding, and thus affect transcription involved in proliferation. In addition, these analyses further indicated that Foxo3a might not be the direct regulator for these downregulated genes and these gene could be indirectly affected in the absence of Tet2. Nonetheless, the new analyses on downregulated genes also confirmed our previous finding on the upregulated genes are specific, as predicted by the reviewer.

How the depletion of Tet2 selectively increases 5hmC level, especially at promoter regions, is an intriguing question. Although intensive efforts have focused on the catalytic activities of Tet proteins to convert 5mC to 5hmC, the Tet proteins can further oxidize 5hmC to its downstream derivatives (5hmC to 5fC/5caC). However, the mechanistic insights in controlling step-wise cytosine demethylation remained largely unexplored. Among all three Tet paralogs in mammals, Tet2 has been implied to be the main player for 5hmC oxidation to downstream derivatives and eventually lead to DNA demethylation (Liu et al, 2016, Nat. Chem. Bio; Xiong et al, 2016, Mol. Cell). This notion has been supported by *in vivo* evidence that depletion of Tet2 promoted 5hmC increase on promoters in mouse embryonic stem cells (Huang et al, 2014, PNAS) and hematopoietic stem cells (Zhang et al, 2016, Nat. Genet.). A very recent study in mESC suggested that another transcription factor, SALL4A,

could directly bind to 5hmC, and coordinate with Tet2 to stimulate further oxidation of 5hmC (from 5hmC to 5fC/5caC) (Xiong et al, 2016, Mol. Cell). SALL4A, in many ways, resembles the characteristics of Foxo3a, such as binding to 5hmC-modified DNA as well as coordinating with Tet2. Thus, it is possible that the Foxo3a possesses similar roles to function as Tet2 co-factor to modulate its enzymatic activity preferentially at the promoter regions. It will be very interesting to pursue these possibilities in the future, but they are beyond the scope of the current study. We have included this point to our discussion.

3. *“The authors performed various sequencing experiments (ChIP-seq, RNA-seq, 5hmC-seq), however they did not validate these hits despite an mRNA q-PCR on three candidate genes (Fig. 6 c-e). This is a necessary control experiment commonly used to confirm the results of genome-wide screens. Likewise, it is also necessary to show the specificity of the antibodies used in the ChIP assay.”*

The reviewer raised a good point. To fully address the reviewer’s comments, we first validated the Foxo3a ChIP-seq experiments in our study. The commercial Foxo3a antibody was specific to detect both endogenous Foxo3a protein at the right size, and effectively immunoprecipitated Foxo3a proteins (Fig. 5b). This antibody also efficiently reacted with commercial Foxo3a recombinant protein used in our new *in vitro* binding assay (Supplementary Fig. 14a). We then downloaded and processed a published Foxo3a ChIP-seq dataset in NSC (Webb et al, 2013, Cell Rep), and overlapped their Foxo3a genome-wide binding sites with our data. We found 56.4% of our Foxo3a ChIP-seq peaks overlapped with published Foxo3a binding sites. It is worthwhile to note that the NSCs presented in Webb et al were derived from whole-mouse forebrains, and they incubated their cells in low growth factor signaling conditions to enhance Foxo3a nuclear accumulation. In comparison, our Foxo3a binding sites were obtained in the native condition from hippocampal NSCs. Nonetheless, the substantial overlapping between our data and theirs ensured the high quality and consistency of our data with published datasets. We have provided a sequencing quality report in the new Supplementary Table 5.

We went on to validate the RNA-seq experiments by qPCR analyses. In our initial submission, we validated three upregulated genes, including Nestin, Kit and Spata13 by qPCR. To address the reviewer’s concern, The expression of several downregulated genes or upregulated genes in *Tet2*^{-/-} aNSCs involved in neuronal differentiation or cell growth/proliferation were further tested by qPCR, and the results were consistent with RNA-seq data (Supplementary Fig. 9a, b). We have provided a sequencing quality report in the new Supplementary Table 5.

The chemical based 5hmC-capture coupled with sequencing (hMe-Seal) has been well established by our group (Song et al, 2011, Nat. Biotechnol.) and extensively validated by us and others. The quality of the hME-Seal was further validated by amplifying the differentially hydroxymethylated regions (DhMRs) in both WT and *Tet2*^{-/-} aNSC using 5hmC-enriched DNA. Six primer sets were designed to target 3 WT DhMRs (The levels of 5hmC in WT were higher than in *Tet2*^{-/-} aNSC identified by hME-Seal) and 3 *Tet2*^{-/-} DhMRs. As expected, all three WT DhMRs displayed significant higher 5hmC levels in WT than *Tet2*^{-/-} aNSC (Supplementary Fig. 11a), and all three *Tet2*^{-/-} DhMRs showed significantly higher 5hmC levels in *Tet2*^{-/-} aNSC (Supplementary Fig. 11b). We have included these information in the revised manuscript.

4. *“It would be useful if the gender information is stated for both in vivo and in vitro studies.”*

We thank the reviewer for this valid point, and have now included this information in the method section of revised manuscript.

5. *“What are the relative expression levels of Tet1, 2, and 3 during adult neurogenesis? And does Tet1 or Tet3 change when Tet2 is manipulated? This is important for understanding the role of Tet2.”*

Since *Tet2*^{-/-} mice were generated by replacing part of exon 3 sequences of the *Tet2* gene with nlacZ/nGFP, the RNA-seq reads of *Tet2* showed only marginal changes (Supplementary Fig. 10b, $p > 0.2$, unpaired *t*-test). Interestingly, endogenous *Tet1* and *Tet3* in *Tet2*^{-/-} aNSCs showed no significant changes, either, supporting that 5hmC dynamics during adult neurogenesis were primarily controlled by *Tet2* (Supplementary Fig. 10b, $p > 0.1$ and $p > 0.6$, unpaired *t*-test). To confirm the predominant roles of *Tet2* in adult neurogenesis, we have performed *Tet1* and *Tet2* double knockdown (dKD) by shRNA in wildtype aNSCs. We found, like *Tet2* single knockout, the *Tet1* and *Tet2* double knockdown in adult NSCs can increase the proliferation and decrease the differentiation. Thus, these new data strongly support *Tet2* plays predominant epigenetic roles in regulating adult neurogenesis.

6. *“There are two aNSC culture conditions referred to in the manuscript - one is proliferating and the other is differentiated aNSC. These two states have different epigenetic status, including 5hmC levels. Though some assays indicate their explicit culture conditions, it is important to define if the aNSCs are differentiated or proliferating in each aNSC culture experiment throughout the manuscript.”*

We thank the reviewer for this important suggestion. We went through the manuscript and clearly defined the culture condition in each experiment. Most the experiments were carried out using proliferating aNSC.

7. *“I will recheck that this was not mentioned and I missed it, but a sequencing quality report should be included on all ChIP-seq, RNA-seq, and 5hmC-seq experiments.”*

As requested by the reviewer, we have provided the sequencing quality report of our ChIP-seq, RNA-seq and 5hmC-seq results (Supplementary Table 5)

8. *“The effectiveness of shRNAs at knocking down the target gene (e.g. Foxo3a, Tet2) should be shown.”*

The shRNA knockdown of both *Foxo3a* and *Tet2* were examined by qPCR and included in the revised manuscript (Supplementary Fig. 5a and Supplementary Fig. 14c).

9. *“Figure 1d is said to show localization of 5hmC in the nuclei of nestin positive aNSCs. But I don't believe the figure is sufficiently clear.”*

We repeated the experiment and included the new data as new Figure 1a-1d. It is consistent with our previous data and clearly demonstrates that 5hmC located in the nuclei of Nestin-positive aNSCs.

10. *“Figure 4b, Please change “upregulated” to “downregulated” on x-axis.”*

We apologize for the error and have corrected it in the revised manuscript.

11. *“Figure 5d: what does the pyramid shape stand for?”*

We apologize for the confusion and have changed the pyramid shape to rectangle in the revised manuscript.

Reviewer #3

We would like to thank this reviewer for the positive comments and constructive suggestions.

1. *“Regarding, Sup Fig. 1, indicate % of Nestin⁺/Sox2⁺ cells among total cells (proliferating condition), and % of Nestin⁺ or Sox2⁺ cell in addition to Tuj1⁺ and GFAP⁺ cells in the differentiation condition. Is there any GFAP⁺/Nestin⁺ (dually positive) cells in the differentiation condition?”*

To answer this good question, we have performed the GFAP and Nestin co-staining in the differentiated condition. There were few GFAP⁺/Nestin⁺ double positive cells in the differentiated conditions, indicating a complete differentiation to the neuronal and glial lineage (Supplementary Fig. 1i-1l).

2. *“Page 7, line 9-11, they described “The aNSCs from the Tet2^{-/-} mice retained the ability to differentiate into neuronal and glial cells (Supplementary Fig. 2c-f)”. However, they actually showed staining of only undifferentiated markers (Nestin and Sox2) in this figure. This is not appropriate.”*

We apologize for the error. We intended to demonstrate the Tet2^{-/-} NSCs still possessed NSC identity with positive Nestin and Sox2 staining here, and the depletion of Tet2 did not result in the loss of aNSC pluripotency and simultaneous differentiation. The impaired neurogenesis in Tet2^{-/-} was demonstrated in main Fig. 2f-2m in the next section. We have revised the text to correct this.

3. *“In terms of Fig. 2, I think it would be better if the authors would provide the staining of Nestin or Sox2 and the quantification of the marker positive cells.”*

We thank the reviewer for this thoughtful suggestion. To address this comment, we performed the Nestin or Sox2 co-staining with BrdU in WT and Tet2^{-/-} NSCs. Tet2^{-/-} NSCs showed substantial increase of proliferation (Supplementary Fig. 3a, b).

4. *“In Sup Fig. 3g and h, they showed both neuronal and glial gene promoters are suppressed by Tet2-KD. These promoters are from genes in opposite lineages. Explain how this could be possible simultaneously.”*

We thank the reviewer for this great question. As shown in Fig. 2f to 2m and supplementary Fig. 1a to 1h, the aNSCs could proliferate and produce both β -III tubulin (TuJ1)- positive neuronal cells and glial fibrillary acidic protein (GFAP)-positive glial cells after two days of differentiation. Thus, our differentiation could simultaneously generate both neuronal and glial cells. At the molecular level, Tet2^{-/-} or Tet2 knockdown affected the expression of genes involved in both lineages, strongly indicating its master roles in generating new neurons and glial cells by shaping the epigenetic landscaping.

5. *“It is not clearly described how the authors come up with the idea of interplay between Tet2 and Foxo3, although both of them have individually known to be important in a variety of biological events.”*

Our initial analyses identified Tet2 as a key regulator in adult neurogenesis. Depletion of Tet2 resulted in the enhanced proliferation and impaired differentiation of adult NSCs. Genome-wide epigenome and transcriptome analyses revealed that loss of Tet2 led to significant alteration of the genome-wide transcriptome landscape related to aNSC proliferation and neuronal lineage commitment. Interestingly, the panel of upregulated genes in the absence of Tet2 appeared to be explicitly associated with proliferating aNSCs, as these genes did not show obvious differences in the differentiated WT or Tet2^{-/-} aNSCs (Fig. 3c, Proliferating versus Differentiation). Thus, we sought to search for transcription regulator that could play direct roles to modulate these gene expression and account for their upregulation in the absence of Tet2. Foxo3a, a mammalian forkhead family member, has been demonstrated to control gene expression during adult neurogenesis (Webb et al, 2013, Cell Rep). Thus, we investigated whether Foxo3a could coordinate with Tet2 in NSC differentiation at the molecular level. We have revised the text to make this point clear.

6. *“As for Fig. 5, they need to show the date for the immunoprecipitation using the control IgG in addition to the data using anti-Foxo3a antibody.”*

We have included a new IP-western blot along with IgG control to clearly demonstrate the association between Foxo3a and Tet2 endogenous proteins in the revised manuscript (Fig. 5b).

7. “Although the authors showed that Tet2 and Foxo3 associate each other, they also indicated that Foxo3a binding to the target genes were increased in Tet2 ^{-/-} aNSCs compared to WT cells. Discuss the mechanism.”

The reviewer raised an excellent question that how Foxo3a will be increasingly recruited to the promoter regions. In fact, there is a positive correlation between increased Foxo3a occupancy and elevated 5hmC in promoters of these upregulated genes in the absence of Tet2 (Supplementary Fig. 12). We speculated this in the discussion of our initial submission that, the presence of Tet2 on these genes could maintain the precise 5hmC homeostasis, which could determine the Foxo3a binding dynamics to keep proper amount of Foxo3a occupancy to control precise gene expression during neurogenesis. Thus, loss-of-Tet2 results in aberrant accumulation of 5hmC, possibly due to the impaired 5hmC oxidation process (5hmC to 5fC/5caC) and recruits additional Foxo3a to ectopically increase the expression of these genes.

To draw the link between 5hmC accumulation and increased Foxo3a occupancy, we first examined the Foxo3a and 5hmC dynamic changes on the promoters of both significantly upregulated and downregulated genes in the absence of Tet2 (Supplementary Fig. 13a). Importantly, both Foxo3a and 5hmC showed concomitantly stronger enrichment on the promoters of upregulated genes than downregulated genes when Tet2 is depleted (Supplementary Fig. 13a). These observations support the notion that the aberrant accumulation of 5hmC upon Tet2 depletion could further recruit substantial more Foxo3a to the promoters of upregulated genes to stimulate their expression. To test the direct interaction between 5hmC and Foxo3a, we synthesized two sets of biotin-labelled DNA oligos with identical sequences based on Foxo3a ChIP-seq top motifs. The cytosines in the consensus sequence were either hydroxymethylated (5hmC probe) or unmodified (control probe), with the latter used as a negative control. Control and 5hmC-containing probes were used for *in vitro* binding by adding with equal amount of Foxo3a recombinant protein. We found that Foxo3a showed significant stronger binding to the 5hmC-modified probes over control *in vitro* (Supplementary Fig. 14a,b). These data, together with Foxo3a/5hmC correlation *in vivo*, revealed the molecular mechanism of upregulated genes that 5hmC accumulation upon Tet2 deletion served as a binding platform to further recruit Foxo3a and stimulate gene expression.

In summary, here we uncovered a positive correlation between significantly increased 5hmC and Foxo3a occupancy specifically on the promoters of these upregulated genes, which explained the mechanism of the upregulation of these genes at the molecular level. These data presented a potentially intriguing negative feedback loop of Tet2-Foxo3a axis. During normal aNSC state, Tet2 interacted with Foxo3a to modulate proper expression level of these proliferation-related genes. While Foxo3a is the direct transcription modulator, Tet2 could actively promote the conversion of promoter 5hmC to its downstream derivatives, which ensure the ideal epigenetic environment to land proper amount of Foxo3a. The absence of Tet2 resulted in the accumulation of 5hmC on the promoters, which could ectopically recruit excessive Foxo3a that led to uncontrolled overexpression of these proliferation related genes (Fig. 7).

8. “Regarding Fig. 4b, “downregulated” (in the text) and “upregulated” (in the figure) are inconsistent. Check which one is correct.”

We apologize for the error and have corrected it in the revised manuscript.

REVIEWERS' COMMENTS:

Reviewer #1 (Remarks to the Author):

The authors have addressed most of my concerns. There are two minor points that remain to be addressed.

Minor points:

1. In Figure 2, the quantification methods of BrdU and other markers should be described in Methods, for example, how many fields was imaged and quantified for each marker.
2. The quality of supplementary Fig. 6a should be improved to make the data more convincing. In the response the reviewers' comments (response to reviewer 1, point 2b), the authors mentioned that Fig. 6a used one-week-old mice. One-week-old mice are not adult mice, therefore, are not the appropriate model for adult neurogenesis. Moreover, the age of the mice used in the experiments should be described in the results and/or methods section of the text.

Reviewer #2 (Remarks to the Author):

The authors have done a good revision to address my prior comments. I do not have further questions.

Reviewer #3 (Remarks to the Author):

As far as I am concerned, the authors addressed all points of criticism raised by the reviewers. The manuscript seems to meet the requirements to stand as a good article for this journal.

Responses to Reviewer #1's Comments:

The authors have addressed most of my concerns. There are two minor points that remain to be addressed.

Minor points:

1. *“In Figure 2, the quantification methods of BrdU and other markers should be described in Methods, for example, how many fields was imaged and quantified for each marker.”*

We thank the reviewer for this suggestion, and have added the quantification methods to the Method section.

2. *“The quality of supplementary Fig. 6a should be improved to make the data more convincing. In the response the reviewers' comments (response to reviewer 1, point 2b), the authors mentioned that Fig. 6a used one-week-old mice. One-week-old mice are not adult mice, therefore, are not the appropriate model for adult neurogenesis. Moreover, the age of the mice used in the experiments should be described in the results and/or methods section of the text.”*

We would like to thank this reviewer for pointing out this error, and apologize for the confusion. The mice used in this figure were 8-week old at the time of BrdU injection, and scarified one week after final BrdU administration. We have included the age of the mice used in the revised manuscript.

Besides the BrdU and Ki67 co-staining, we did provide the results from many other additional experiments to demonstrate the increased BrdU staining in Tet2^{-/-} aNSCs (Fig. 2a-2d, Fig. 6f and Supplementary Fig. 3a-3b, Supplementary Fig. 6b), Tet2 siRNA knockdown aNSCs (Supplementary Fig. 6f), SVZ *in vivo* (Supplementary Fig. 7), along with increased expression of proliferation markers (Fig. 3, Fig. 6c, Supplementary Fig. 4b, Supplementary Fig.5b). These data together provide strong evidence for our conclusions.

The BrdU and Ki67 co-staining *in vivo* has been a technical challenge for us. Given the reviewer's request, we have repeated the BrdU and Ki67 co-staining. Our new image is consistent with our previous observations that BrdU⁺/Ki67⁺ positive cells increased *in vivo*. We have included the new image in the revised manuscript (Supplementary Figure 6a).